# EvA: An Evidence-First Audio Understanding Paradigm for LALMs

## Abstract

While Large Audio Language Models (LALMs) have demonstrated remarkable capabilities in audio understanding tasks, their performance degrades sharply in complex acoustic scenes, revealing a fundamental limitation in their perceptual grounding. In this work, we first identify a critical failure mode that exposes this limitation: state-of-the-art LALMs paradoxically struggle more with simple evidence-extraction tasks than with complex reasoning ones. We diagnose this as a breakdown in acoustic evidence grounding, a problem rooted in systemic information loss during feature encoding and fusion. To address this, we introduce EvA (Evidence-First Audio), a new paradigm that prioritizes maximizing the fidelity of acoustic evidence. EvA's dual-encoder architecture combines Whisper with CED-Base, a ViT-based general audio encoder, and pioneers a structure-preserving, two-stage fusion process. First, it enriches evidence by hierarchically aggregating multi-level features from within the CED-Base encoder. Second, it integrates this representation with Whisper's output via a time-aligned, inject-and-add mechanism that guarantees perfect temporal integrity. To facilitate training for this paradigm, we co-develop EvA-Perception, a large-scale open-source dataset with high-temporal-precision annotations. Our resulting model establishes a new open-source state-of-the-art on multiple challenging benchmarks, including MMAU, MMAR, and MMSU. Crucially, EvA achieves its most significant gains on perception-heavy subsets, validating our hypothesis that addressing the evidence bottleneck is key to unlocking the next level of audio understanding.

## 1 Introduction

Large Audio Language Models (LALMs) aim to empower machines with the ability to listen, understand, and reason from sound. While recent systems like Qwen2.5-Omni (Xu et al., 2025) and Kimi-Audio (Ding et al., 2025) have demonstrated impressive performance on various benchmarks, their capabilities degrade sharply when confronted with complex acoustic scenes involving overlapping events, faint signals, or fine-grained temporal cues.

We argue this fragility stems not from a deficit in reasoning but from a more fundamental breakdown in acoustic perception. As illustrated in Figure 1, the performance gap between leading LALMs and humans is larger for perception-centric tasks than for reasoning-centric ones. On the MMSU benchmark, the model-human gap in perception is a stark 48.4 points (42.8% vs. 91.2%), whereas the reasoning gap is a comparatively narrow 13.3 points. This disparity reveals what we term the "evidence bottleneck": the primary performance ceiling is not a model's cognitive capacity, but its foundational inability to accurately perceive and represent acoustic evidence for subsequent reasoning.

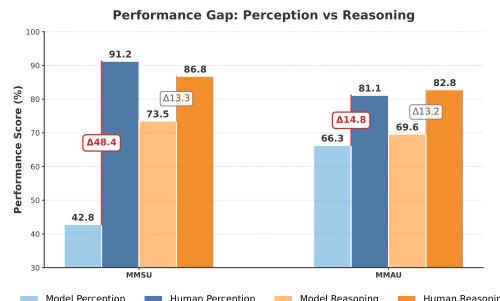

Figure 1: Perception–reasoning performance gap comparison between model and human. Model performance are averaged over Qwen2.5–Omni and Kimi–Audio–7B.

We posit this "evidence bottleneck" is a systemic limitation rooted in information-theoretic principles, arising from three flaws in the prevailing LALM design. First, **reasoning-centric optimization**: Post-training methods like SFT and GRPO refine textual reasoning but are fundamentally incapable of correcting upstream perception errors. Once acoustic details are lost in the initial encoding stages, no downstream process can recover them. This places a hard ceiling on performance, irrespective of the reasoning model's power. Second, **frequency information attrition**: Dominant encoders process spectrograms as 1D sequences, collapsing the frequency dimension and discarding crucial spectral cues. Third, **flawed alignment interfaces** that manifest in two modes: lossy temporal compression via modules like Q-Formers (Tang et al., 2023), or misaligned feature concatenation that provides no unified temporal coordinate for the LLM to exploit (Ghosh et al., 2024).

To systematically dismantle this bottleneck, we introduce **EvA (Evidence-First Audio)**, an architectural paradigm designed for **maximal evidence preservation**. EvA processes audio through two parallel, complementary streams: a Whisper encoder for phonetic content and a generalist audio encoder (CED-Base (Dinkel et al., 2024)) for the rich tapestry of non-speech sounds. Its core innovation is a two-stage, non-compressive fusion process that directly counteracts the information loss identified above. First, it performs **hierarchical evidence aggregation** within the generalist encoder, fusing features from intermediate layers to create a multi-scale representation that retains both local and global acoustic details. Second, this enriched acoustic sequence is integrated with the speech features via a **time-aware additive fusion**, which preserves the full temporal resolution of both streams and provides the LLM with a unified, high-fidelity timeline of all acoustic events.

To realize this architecture's potential, we develop EvA-Perception, a large-scale audio QA dataset of 54K high-fidelity captions and 500K QA pairs. Generated using temporal annotations from AudioSet-Strong (Hershey et al., 2021), it is explicitly designed to train models to ground reasoning in fine-grained acoustic evidence. Our model, fine-tuned on only 380 hours of audio, establishes a new open-source state-of-the-art on complex audio understanding, acoustic scene and event classification benchmarks, including MMAU (Sakshi et al., 2024), MMAR (Ma et al., 2025), MMSU (Wang et al., 2025), CochlScene(Jeong & Park, 2022), TUT2017(Mesaros et al., 2016) and Vocal-Sound(Gong et al., 2022). Crucially, as hypothesized, the largest performance gains occur on the most perception-heavy tasks, directly validating our evidence-first approach.

Our main contributions are summarized as follows: **Problem Diagnosis:** We identify and empirically validate the "evidence bottleneck" in SOTA LALMs, arguing from information-theoretic principles that poor acoustic grounding, not reasoning, is the primary performance limiter. **The EvA Architecture:** We propose EvA, a dual-stream paradigm for evidence preservation, featuring a novel, non-compressive fusion process that hierarchically aggregates multi-level acoustic features and integrates them while maintaining full temporal fidelity. **Open-source Dataset and SOTA Model:** We release EvA-Perception, a high-temporal-precision dataset for perceptual training, and the EvA model, which sets a new open-source state-of-the-art on multiple complex audio understanding benchmarks.

## 2 RELATED WORKS

### 2.1 LARGE AUDIO LANGUAGE MODELS

The field of Large Audio Language Models (LALMs) has recently seen rapid progress, with systems like Qwen2-Audio (Chu et al., 2024), Qwen2.5-Omni (Xu et al., 2025), and Kimi-Audio (Ding et al., 2025) demonstrating strong performance on various audio understanding tasks. A common thread among these models is their reliance on a single, ASR-centric Whisper encoder (Radford et al., 2023). While this foundation is powerful for speech, extending its perceptual capabilities to general non-linguistic audio often incurs prohibitive training costs and risks degradation of its original ASR distribution. Recognizing this trade-off, earlier work explored multi-stream architectures. SALMONN (Tang et al., 2023), for instance, pioneered a dual-encoder approach but was hampered by its fusion strategy, which employs a Q-Former for lossy temporal compression, creating a critical information bottleneck. This compression effect can be directly observed in the reduced audio token sequence length(Appendix A.10). Subsequent work like GAMA (Ghosh et al., 2024) explored hierarchical feature aggregation, yet fell into a different trap. It structurally corrupted its own temporal representation by concatenating the full feature sequence from its aggregator with a few context-less tokens from a Q-Former. Both approaches, despite their initial promise, ultimately failed to deliver

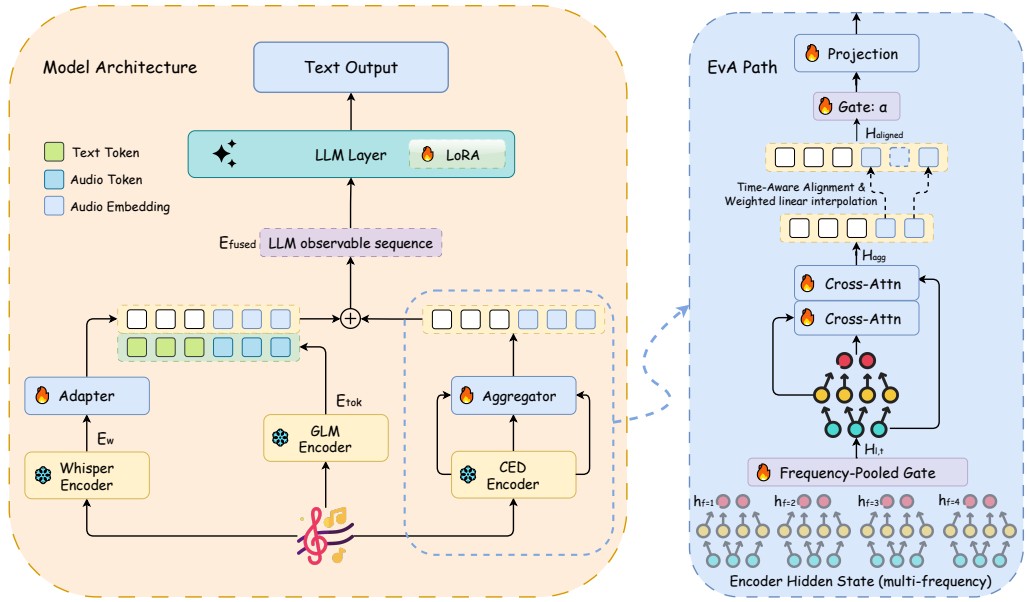

Figure 2: **Model architecture.** The left half of the left panel shows the Kimi-Audio backbone, while the right half illustrates the additional EvA Path modules. Audio is encoded into four frequency-band features by the CED Encoder, pooled across frequencies, fused via cross-attention, temporally aligned with Whisper, and integrated through gated additive fusion.

high-fidelity, temporally coherent evidence to the LLM. Our work learns from these precursors, retaining the concept of rich evidence while introducing a novel fusion mechanism that avoids both temporal compression and structural corruption.

## 2.2 AUDIO UNDERSTANDING & REASONING BENCHMARKS

Audio-language evaluation has evolved from foundational captioning tasks, such as on AUDIOCAPS (Kim et al., 2019) and CLOTHO (Drossos et al., 2020), to more complex reasoning-centric benchmarks. Datasets like MMAU (Sakshi et al., 2024), MMAR (Ma et al., 2025), and MMSU (Wang et al., 2025) are specifically designed to probe deeper abilities, such as localizing events in time, inferring causality, and reasoning about the complex interplay of non-linguistic sounds. It is precisely on these demanding benchmarks that the architectural trade-offs of current LALMs are magnified. This reveals a critical insight: while advanced reasoning policies are undoubtedly beneficial, the ultimate performance ceiling on these tasks is often imposed by the fidelity of the initial perception. A model cannot reason about acoustic details it has failed to accurately perceive. This observation pinpoints a critical need for a paradigm that prioritizes the fidelity of perceptual input, a need that directly motivates our "Evidence-First" approach and the development of our EvA-Perception dataset.

## 3 A PRINCIPLED VIEW ON THE EVIDENCE BOTTLENECK

The perception–reasoning gap observed in Sec. 1 suggests a fundamental architectural limitation, not merely a training deficit. In this section, we view the LALM pipeline through a *conceptual, architecture-level information-flow lens*, modeling how acoustic evidence may be transformed and attenuated along the processing chain, so as to find out an efficient way to enhance performance. We first discuss the informational constraint of single–path models (Sec. 3.1), then explain why a dual–path architecture can relax this constraint in practice (Sec. 3.2). Formal analysis is deferred to the appendix A.2; here we keep the statements at a high level to guide design.

**Notation.** We define the following variables: $Z$ — **the latent acoustic evidence** required to answer the downstream task(e.g., event identities, temporal boundaries and ordering); $X$ — the raw audio waveform; $H$ — the encoder's hidden representation, $O$ — the final representation passed to the language model, and $Y$ — the model's textual output. Mutual information between any two variables is denoted as $I(\cdot;\cdot)$. We only assume that a joint distribution over $(Z, X)$ exists; this minimal assumption suffices for *qualitative* reasoning about information flow. Mutual information is denoted $I(\cdot;\cdot)$.

### 3.1 SINGLE-PATH LALMS: AN INFORMATION CONSTRAINT

Conditioned on fixed parameters after training, the inference-time forward pass is a composition of *deterministic* mappings of $X$:

$$H = E_\theta(X), \qquad O = P_\theta(H), \qquad Y = \pi_\theta(O).$$

Under this setting, the data-processing inequality (DPI) for deterministic functions implies

$$I(Z;Y) \ \leq \ I(Z;O) \ \leq \ I(Z;H) \ \leq \ I(Z;X). \tag{1}$$

We use Eq. equation 1 *conceptually* to compare *relative* information retention across stages/paths: post-training that optimizes only the policy $\pi_\theta$ (e.g., SFT/GRPO(Group Relative Policy Optimization)) can improve how well the model *exploits* evidence already present in $O$, but it does not *restore* acoustic details attenuated upstream in $X \to H \to O$. This observation motivates architectural choices that prioritize evidence retention before the LLM (see Sec. 3.3). A derivation of equation 1 under fixed parameters appear in Appendix A.2.

### 3.2 THE ADVANTAGE OF DUAL PATHS: POSSIBLE MORE INFORMATION

Consider two complementary perception paths over the same input $X$: a speech-centric path producing $O_1 = P_1(E_1(X))$ (e.g., Whisper) and a general-audio path producing $O_2 = P_2(E_2(X))$ (e.g., CED-Base). The LLM receives the joint observation $(O_1, O_2)$.

By the chain rule,

$$I(Z;O_1,O_2) \ = \ I(Z;O_1) + I(Z;O_2 \,|\, O_1) \ \geq \ I(Z;O_1).$$

We cite this identity to articulate the *complementarity intuition*: when the second path contributes cues not already captured by the first, the joint observation is—in a qualitative sense—no less informative than either stream alone. We do not estimate $I(\cdot\,;\cdot)$ nor claim quantitative increases or bounds in the main text; empirical ablations in Table 3 align with this intuition.

### 3.3 IMPLICATIONS FOR LALM DESIGN

This perspective leads to two practical guidelines:

**(i) Prioritize the perceptual front-end.** The primary bottleneck often lies in evidence retention through the encoder/fusion stack rather than in the LLM policy; improving upstream access to fine-grained cues is therefore crucial.

**(ii) Favor time-aligned, non-compressive fusion.** Fusion interfaces that preserve temporal resolution and avoid heavy compression are aimed at minimizing avoidable information loss. In our system, the general-audio stream is aligned to the speech timeline and injected via an add-based mechanism with a learnable gate, which is structure-preserving and sequence-length neutral.

## 4 METHOD: EVIDENCE-FIRST AUDIO UNDERSTANDING PARADIGM

### 4.1 ARCHITECTURE

EvA adopts a dual-path architecture on top of the **Kimi-Audio-7B** backbone that delivers time-aligned, structure-preserving acoustic evidence. As shown in Figure 2, the raw waveform is encoded by two complementary encoders, a speech-centric Whisper path and a generalist CED path. Their evidence is aligned to the token timeline and injected into the backbone LLM input space without changing sequence length. Details on initialization and the freezing/training policy are deferred to Sec. 4.3.

**Complementary Dual Encoders.** We employ two encoders that capture distinct but complementary information channels. (i) The **Whisper encoder** ($E_W$) is pre-trained on large-scale ASR corpora and excels at extracting robust, high-level linguistic features and provides a strong foundation for speech-related tasks. (ii) The **CED-Base encoder** ($E_C$), a Vision Transformer (ViT) based model trained for general audio event recognition, exposes richer non-linguistic cues (background events, music, transients). We extract hidden states from its shallow, middle, and final layers to harvest multi-scale semantics. These two encoders provide complementary views of the acoustic scene, corresponding to the distinct information channels $O_1$ and $O_2$ in our information-theoretic analysis, forming a comprehensive perceptual basis for the downstream LLM.

**Hierarchical Evidence Aggregation.** Standard encoders, which only expose their final-layer features, create an internal information bottleneck long before the LLM. The Data Processing Inequality dictates that these final features cannot be more informative than the collection of intermediate representations. To mitigate this loss, we introduce a hierarchical aggregation process that fuses and harvests features across the frequency domain and from multiple network depths.

First, in the **frequency domain**, we leverage the fact that the ViT-based CED encoder's feature maps implicitly retain a frequency axis. For the raw feature maps $\tilde{\mathbf{h}}_l \in \mathbb{R}^{B \times T \times F \times D_c}$ extracted from layer $l \in \{4, 8, L\}$ (where $F$ is the number of frequency bands), we apply a lightweight gated attention mechanism. This performs a learnable, weighted pooling across the frequency bands for each time step:

$$\mathbf{h}_{l,t} = \sum_{f=1}^{F} \text{softmax}(\text{gate}(\tilde{\mathbf{h}}_{l,t,f})) \cdot \tilde{\mathbf{h}}_{l,t,f} \tag{2}$$

This operation dynamically focuses on the most salient frequency bands at each moment, compressing the 2D feature map into a more informative 1D temporal sequence, which we denote as $\mathbf{H}_l \in \mathbb{R}^{B \times T \times D_c}$.

Second, in the **cross-layer domain**, we fuse these frequency-aggregated features, $\mathbf{H}_4, \mathbf{H}_8,$ and $\mathbf{H}_L$, using a two-stage cascaded cross-attention mechanism implemented in our *Aggregator*. It first enriches the high-level semantic features $\mathbf{H}_L$ with mid-level temporal details from $\mathbf{H}_8$, and then grounds the resulting representation with low-level acoustic patterns from $\mathbf{H}_4$:

$$\mathbf{H}' = \text{LayerNorm}(\text{CrossAttn}(\text{Q=}\mathbf{H}_L; \text{K}, \text{V=}\mathbf{H}_8) + \mathbf{H}_L) \tag{3}$$

$$\mathbf{H}_{\text{agg}} = \text{LayerNorm}(\text{CrossAttn}(\text{Q=}\mathbf{H}'; \text{K}, \text{V=}\mathbf{H}_4) + \mathbf{H}') \tag{4}$$

This cascaded, two-stage aggregation process produces a informative feature sequence $\mathbf{H}_{\text{agg}}$ that embodies acoustic evidence integrated across both multiple frequency bands and multiple levels of abstraction.

**Time-Aware Alignment and Inject-and-Add Fusion.** The final and most critical step is to integrate the evidence from both encoder paths without creating a fusion bottleneck. A key challenge is the temporal mismatch: the LLM-aligned Whisper features have a stride of **80 ms**, whereas the effective stride of our aggregated CED features $\mathbf{H}_{\text{agg}}$ is coarser at **160 ms**. To reconcile this, we upsample the CED evidence onto the Whisper timeline using a **time-aware linear interpolation**. This method respects the true mel-frame timestamps of each feature window. For each Whisper token's timestamp, we identify its two nearest neighbors in the CED sequence and compute a weighted average, carefully accounting for the temporal coverage of each feature window to avoid phase drift and preserve transients. The full algorithm is detailed in Appendix A.3. This process yields a temporally-aligned CED feature sequence, $\mathbf{H}_{\text{aligned}} \in \mathbb{R}^{B \times T_w \times D_c}$, that now shares the same timeline as the Whisper features.

Both the original Whisper features $\mathbf{E}_W$ and the aligned CED features $\mathbf{H}_{\text{aligned}}$ are then passed through separate lightweight projection heads ($\text{Proj}_W$ and $\text{Proj}_C$ respectively) to map them to the LLM's hidden dimension. Finally, they are integrated using our **inject-and-add** strategy. We chose this approach for three key principles: **(1) Efficiency:** simple vector addition incurs minimal computational overhead. **(2) Structural Compatibility:** it preserves the original sequence length and causality, requiring no modification to the LLM backbone. **(3) Controllability:** it allows for stable training via a learnable gate.

Table 1: Comparison of open-source audio caption datasets.

| Name | # of Audio/QA | Avg. Caps Len | Visual | Music | Speech | Integration | Temporal Info |
|------|---------------|---------------|--------|-------|--------|-------------|---------------|
| AudioCaps (Kim et al., 2019) | 46k/46k | 9.03 | ✗ | ✗ | ✗ | ✗ | ✗ |
| Clotho (Drossos et al., 2020) | 5k/5k | 11.00 | ✗ | ✗ | ✗ | ✗ | ✗ |
| LAION-Audio-630K (Wu et al., 2023) | 630k/630k | 7.30 | ✗ | ✗ | ✗ | ✗ | ✗ |
| WavCaps (Mei et al., 2024) | 403k/403k | 7.80 | ✗ | ✗ | ✗ | ✗ | ✗ |
| AudioSetCaps (Bai et al., 2025a) | 1.9M/1.9M | 28.00 | ✗ | ✗ | ✗ | ✗ | ✗ |
| Auto-ACD (Sun et al., 2024) | 1.5M/1.5M | 18.10 | ✓ | ✗ | ✗ | ✓ | ✗ |
| CompA-R (Ghosh et al., 2024) | 62k/200k | 18.00 | ✓ | ✗ | ✗ | ✓ | ✗ |
| FusionAudio-1.2M (Chen et al., 2025) | 1.2M/6M | 47.18 | ✓ | ✓ | ✓ | ✓ | ✗ |
| **EvA-Caps/QA** | 54K/500K | **67.99** | ✓ | ✓ | ✓ | ✓ | ✓ |

The final fused embedding $\mathbf{E}_{\text{fused}}$ is computed based on a mask $\mathbf{M}$ that identifies audio token positions:

$$\mathbf{E}_{\text{fused}}[i] = \begin{cases} \left(\mathbf{E}_{\text{tok}}[i] + \text{Proj}_W(\mathbf{E}_W[i])\right) \cdot \sqrt{2} + \alpha \cdot \text{Proj}_C(\mathbf{H}_{\text{aligned}}[i]), & \text{if } \mathbf{M}[i] = 1 \\ \mathbf{E}_{\text{tok}}[i], & \text{if } \mathbf{M}[i] = 0 \end{cases} \quad (5)$$

where $\mathbf{E}_{\text{tok}}$ are the initial token embeddings, and $\alpha$ is a learnable scalar gate initialized to a small value. This allows the model to gradually incorporate the generalist evidence without perturbing the LLM's pre-trained knowledge during early training stages. This strategy enriches each audio token locally, thereby circumventing the information bottlenecks typical of heavy, compressive fusion modules.

## 4.2 EvA-Perception: A Dataset for Evidence-Grounded Training

**Motivation** Perception-heavy failures in LALMs often stem from weakly grounded supervision: generic audio-caption lack temporal information or fine-grained high-fidelity cues (as shown in Table 1). **EvA-Captions**, a core component of EvA-Perception, couples temporal-order evidence with instruction-style supervision, guiding models to extract diverse evidence with temporal information from high-precision manual labels and then reason to mitigate bias.

**Data Construction** We build diverse complementary resources via a multi-expert pipeline (details and prompts in Appendix A.5). *AudioSet-Strong* (Hershey et al., 2021) provides audio resource and time-localized manual soft labels as acoustic priors, then *Gemini-2.5-Pro* (Comanici et al., 2025) transforms them into event-ordered, natural-language captions; *Whisper* (Radford et al., 2023) contributes ASR for speech presence; *OpenMu* (Zhao et al., 2024) adds music-oriented details; *Qwen-2.5-VL-72B* (Bai et al., 2025b) provides visual cues used exclusively to disambiguate underspecified audio events; and *QwQ-32B* (Team, 2025) consolidates all descriptions into a single fine-grained coherent caption while preserving temporal order.

**Results** This pipeline yields ∼**54K** fine-grained captions (**150 h**). From each caption we automatically derive QA pairs spanning closed-ended and open-ended queries at a 2:3 ratio (see Appendix A.5 for detail), producing ∼**500K** items. As a result, we construct:

1. **EvA-Captions & EvA-QA**: consolidated, event-ordered, fine-grained captions (∼54K / 150 h) and corresponding QA pairs (∼500K; closed/open 2:3).

2. **EvA-Alignment & EvA-Perception**: aggregated datasets, where EvA-Alignment (including EvA-Captions and diverse resources) is used for encoder alignment, and EvA-Perception (including EvA-Captions, EvA-QA, and broader resources) is used for SFT. The specific composition of these datasets can be seen in Appendix A.5.

## 4.3 Training Strategy

We perform a two-stage training strategy to first stably integrate our fusion module with the pretrained backbone, and then to fine-tune the entire system for complex, evidence-grounded reasoning.

**Backbone Initialization.** We build EvA on the public **Kimi-Audio-7B** backbone, keeping the pretrained Whisper encoder and CED-Base frozen. This avoids costly re-pretraining, preserves

Table 2: Main results on benchmarks. EvA achieves open-source SOTA with particularly strong gains on *Perception*. All numbers are from our unified reproduction under 0-shot setting. The exact definition of perception and reasoning in benchmarks and reproduction setting can be seen in Sec. 5.1.

| Model | Arch. | MMAU | | MMAR | | MMSU | | CochlScene | TUT | VocalSound |
|-------|-------|------|------|------|------|------|------|------------|-----|------------|
| | | Perc. | Reas. | Perc. | Reas. | Perc. | Reas. | | | |
| GPT-4o Audio | **Closed Source** | 60.68 | 59.94 | 51.12 | 60.69 | 43.25 | 75.04 | 36.50 | 15.56 | 79.70 |
| Qwen2-Audio | Single Enc. | 49.85 | 50.52 | 41.38 | 47.56 | 41.85 | 69.88 | 36.50 | 28.55 | 86.47 |
| Qwen2.5-Omni | Single Enc. | 67.18 | 70.16 | 54.81 | 61.12 | 42.25 | 74.13 | 49.49 | 49.83 | 90.39 |
| Kimi-Audio | Single Enc. | 65.33 | 69.10 | 49.21 | 60.40 | 43.36 | 72.80 | 48.08 | 41.97 | 91.87 |
| Audio-Reasoner | Single Enc.+CoT | 69.35 | 62.92 | 47.21 | 49.91 | 44.45 | 73.95 | – | – | – |
| R1-AQA | Single Enc.+RL | 71.83 | 66.91 | 53.69 | 51.26 | 42.13 | 69.92 | – | – | – |
| **EvA(Ours)** | **EvA(Dual Enc.)** | **80.19** | **71.79** | **55.26** | **62.59** | **47.44** | **74.57** | **74.94** | **66.24** | **93.48** |

encoder distributions and tokenization, and ensures fair comparison by isolating the effect of our evidence-first fusion.

**Stage 1: Alignment Tuning.** In this stage, only the newly introduced modules (the CED Aggregator and projection heads) are trained, using next-token cross-entropy on text tokens. The objective is to learn the mapping from the generalist audio encoder's feature space to the LLM's input embedding space, without disrupting the model's pre-trained weights. The small initialization of the gate $\alpha$ in Eq. 5 is critical for stability.

**Stage 2: Instruction Fine-Tuning.** We fully train the CED Aggregator and the Whisper adapter; the LLM backbone is updated via **LoRA**, while both encoders stay frozen. We continue to use the same text-only cross-entropy objective on training set.

## 5 EXPERIMENTS

In this section, we evaluate our method on the different benchmarks. Beyond the official leaderboards, we also report results under unified *Perception* and *Reasoning* splits to better reflect the effect of our method. EvA achieves new open-source SOTA across three main benchmarks, with the largest gains on *Perception* subsets, consistent with our evidence-first design.

### 5.1 EXPERIMENTAL SETUP

**Benchmarks.** We evaluate mainly on three audio understanding benchmarks—**MMAU**, **MMAR**, and **MMSU**—and three acoustic scene and event classification benchmarks—CochlScene, TUT2017 and VocalSound that together span perceptual coverage, domain specialization, and deep reasoning. To directly test our central hypothesis regarding the evidence bottleneck, we categorize the sub-tasks of each benchmark into two primary axes: **Perception** and **Reasoning**. This categorization allows us to quantify performance gains specifically on tasks that depend on acoustic perception versus those that test abstract reasoning, the detailed categorization can be found in Appendix A.7. For completeness, we also report results under each benchmark's original categories and describe our special handling of answer ordering in *MMAU* in Appendix A.7.

**Baselines.** We compare to strong general understanding systems and reasoning systems: *GPT-4o-Audio*, *Qwen2-Audio* (Chu et al., 2024), *Qwen2.5-Omni* (Xu et al., 2025), *Kimi-Audio* (Ding et al., 2025), *Audio-Reasoner* (Xie et al., 2025b) and *R1-AQA* (Li et al., 2025). All baselines are run under a unified inference protocol, and numbers reported are from our reproduction to ensure fairness.

**Implementation Details.** Training follows the procedure outlined in Sec. 4.3. In **Stage 1 (Alignment)**, we use EvA-Alignment for alignment, training only the CED Aggregator with all other encoders frozen. We use a learning rate of $1\times10^{-3}$, train for 5 epochs, with a per-device batch size of 2 and gradient accumulation of 8, resulting in a global batch size of 128. In **Stage 2 (SFT)**, EvA-Perception is used for instruction tuning. Both the CED Aggregator and the Whisper adapter are trained, while the LLM backbone is fine-tuned via **LoRA** with a rank of 64, $\alpha = 64$, and a dropout

Table 3: Ablations of the EvA fusion path. On *AudioCaps*, we use the CLAP encoder (Wu et al., 2023) to embed text/audio; **Cos** is cosine similarity, and higher is better for all CLAP metrics. On MMAU/MMAR/MMSU, we report *Perception* accuracy (Sec. 5.1). *Adapter* denotes the Whisper adapter; *LLM* denotes LoRA on the backbone; "*mask CED path*" disables the CED branch at fusion.

| Stage | Setting | Trainables | Start Ckpt | AudioCaps (CLAP) | | |
|-------|---------|------------|------------|------|------|------|
| | | | | Cos | R@1 | R@5 |
| S0 | Base Model | N/A | N/A | 14.61 | 9.27 | 24.97 |
| S1(1) | w/o CED path | Adapter | S0 | 35.40 | 18.50 | 43.60 |
| S1(2) | mask CED path in inf | Adapter & CED Agg. | S0 | 34.37 | 17.76 | 41.44 |
| S1(3) | w/o frequency pooling | Adapter & CED Agg. | S0 | 35.54 | 21.24 | 49.61 |
| S1(4) | w/o crossing fusion | Adapter & CED Agg. | S0 | 28.63 | 11.82 | 29.74 |
| S1(5) | Q-former | Adapter & CED Q-former | S0 | 36.24 | 20.08 | 47.36 |
| S1(0) | EvA | Adapter & CED Agg. | S0 | **36.77** | **22.77** | **49.81** |

| Stage | Setting | Trainables | Start Ckpt | MMAU | MMAR | MMSU |
|-------|---------|------------|------------|------|------|------|
| S0 | Base Model | N/A | N/A | 65.33 | 49.21 | 43.36 |
| S2(1) | mask CED path in inf | Adapter & LLM | S1(0) | 75.85 | 54.59 | **48.18** |
| S2(0) | EvA | Adapter & CED Agg. & LLM | S1(0) | **80.19** | **55.26** | 47.44 |

rate of 0.05. The learning rate is reduced to $5 \times 10^{-5}$ with 2 epochs, a per-device batch size of 2, and gradient accumulation of 16, yielding a global batch size of 256. The maximum sequence length for training and inference is set to 1024. Both training and decoding use greedy sampling (temperature 0) with a max length of 1024. Each stage takes approximately 12 hours to complete on $8 \times$A100 GPUs. For the detail of training setting, please refer to Appendix A.4.

## 5.2 MAIN RESULTS

In this section, we analyze the performance of EvA on the benchmarks. As shown in Table 2, EvA achieves open-source state-of-the-art (SOTA) performance on both *Perception* and *Reasoning* tasks, with a particularly strong improvement in *Perception* compared to its base model, Kimi-Audio. We also conduct qualitative analysis in Appendix A.9

**Overall Performance Comparison.** EvA sets a new state-of-the-art for open-source models across former three benchmarks. As shown in Table 2, our model attains an average score of **74.63** on MMAU, **59.30** on MMAR, and **61.28** on MMSU. This represents a substantial improvement over all prior methods, including a +7.4 point gain on MMAU over its own Kimi-Audio base model, demonstrating the effectiveness of our architectural modifications. In later three benchmarks, EvA achieves high progress compared to base model, Kimi-Audio, with over +25 point.

**A Remarkable Progress in Perception.** The most notable enhancement is seen in tasks which highly lies on *Perception*, such as Perception subfields on MMAU/MMAR/MMSU and acoustic scene/event classification, where EvA achieves a **14%** improvement on MMAU compared to its base model. This demonstrates the effectiveness of the dual-encoder and evidence-first fusion approach, which significantly enhances the model's ability to capture fine-grained audio details. This progress is also evident in MMAR and MMSU. In contrast, the improvement in *Reasoning* is smaller but still significant, showcasing the versatility of the model across tasks in different levels.

**Performance Analysis.** The dramatic improvement in *Perception* underscores the success of the EvA architecture. By leveraging the strengths of both the Whisper encoder for speech-related tasks and the CED encoder for general audio recognition, EvA excels at extracting and fusing audio evidence with greater precision. The **evidence-first audio** paradigm plays a crucial role in preserving fine acoustic details, which is essential for tasks requiring high accuracy in audio understanding.

## 5.3 ABLATION STUDIES

**Setup.** We study the proposed CED-based fusion path from two perspectives. First, we assess its **overall contribution** by comparing variants that (do not) use the CED branch during *alignment* (Stage 1) and *perception SFT* (Stage 2). Second, we analyze the necessity of **internal design** of

the CED Aggregator, including the frequency-gated pooling over bands and the top–down cross-layer fusion across CED layers. Stage 1 variants are trained on **EvA-Alignment** and evaluated on *AudioCaps* using CLAP metrics, while Stage 2 variants are trained on **EvA-Perception** from the same aligned checkpoint and evaluated on MMAU, MMAR, and MMSU.

**Overall effect of the CED branch.** Compared to the S0 base model without any CED path, enabling the CED Aggregator during Stage 1 alignment substantially improves CLAP retrieval on *AudioCaps*: all Stage 1 variants that use the CED branch outperform the S0 backbone, while masking the CED path at inference time yields consistently worse scores than using it throughout (Table 3, top). Starting from the same aligned checkpoint, keeping the CED stream active in Stage 2 perception SFT further boosts MMAU and MMAR over masking the CED path (Table 3, bottom), indicating that the CED encoder contributes complementary non-speech evidence that remains useful after alignment. The speech-heavy MMSU benchmark shows a mild reversal, which is consistent with our focus on general audio rather than speech-specialized training.

**Aggregator design: frequency pooling and cross-layer fusion.** Within the CED branch, we next ablate the Aggregator design (Table 3, top). Removing the top–down cross-layer fusion and using only the last CED layer feature (S1(4)) leads to a marked drop in all CLAP metrics with a comparable parameter budget. This directly supports our claim that intermediate CED layers supply complementary acoustic information that is lost when relying solely on the final layer. Replacing our hierarchical Aggregator with a window-level Q-Former (S1(5)), similar in spirit to SALMONN, improves over weaker baselines but still underperforms the full EvA configuration (S1(0)), indicating that non-compressive, multi-level fusion is more effective than aggressive window compression in our setting. We also analyze the structural shortcoming of Q-Former module simply compared with Aggregator in Appendix A.10.

We further examine the frequency-gated pooling over four sub-bands. Simply averaging over bands (S1(3)) instead of using a learnable gate consistently degrades CLAP scores, showing that the gated pooling helps the model reweight spectral regions. To probe which bands matter, we perform a coarse band-masking diagnostic by zeroing or keeping individual bands within the same gate (Appendix A.8). Across the benchmarks, masking any single band or keeping only one band is always worse than using the full range, and no individual band dominates across all tasks.

Taken together, these ablations suggest that (i) EvA benefits from multi-layer CED features rather than only the last layer, and (ii) the four sub-bands provide complementary cues, making EvA a reasonable and robust design choice rather than a rough module assembly.

## 6 Conclusion

In this work, we identified the evidence bottleneck as a critical limitation in Large Audio Language Models (LALMs), where perceptual grounding fails in complex acoustic scenes. To address this, we introduced EvA, an evidence-first paradigm that combines a dual-encoder architecture with a novel, non-compressive and temporally faithful fusion mechanism to preserve acoustic evidence. Our approach, supported by the EvA-Perception dataset, achieves state-of-the-art performance on benchmarks like MMAU, MMAR, and MMSU, with significant gains in perception-heavy tasks, validating the importance of prioritizing evidence fidelity for advanced audio understanding.

**Limitations** While EvA advances evidence-grounded audio understanding, several limitations remain: our curated audio–text training corpus currently uses English-only captions, even though the paired audio and evaluation benchmarks contain multilingual inputs, so a more systematic multilingual textual supervision and evaluation is still missing; temporal reasoning is constrained by soft event boundaries from AudioSet-Strong, and music analysis lacks expert-level concepts like pitch or harmony. Addressing these challenges represents key directions for future work.

ETHICS STATEMENT

We used only publicly available, anonymized datasets whose licenses allow academic research. No personal or sensitive information was collected. The study did not involve human subjects and therefore did not require IRB approval. The model could potentially be misused to generate misleading text. We will mitigate this risk by releasing it under a responsible-use license that explicitly prohibits harmful applications. There are no competing financial interests. All experiments were conducted in compliance with local data-protection laws.

REPRODUCIBILITY STATEMENT

To ensure full reproducibility, we will open-source the implementation code, model weights, and datasets used in this study. All experimental details and hyperparameters are thoroughly documented in the paper.

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

# A  APPENDIX

## A.1  USE OF LLMS

Large language models were utilized for specific tasks, such as assisting with coding and providing grammar checks and language refinement during the writing of this paper. All scientific content, including research design, experimentation, data analysis, and conclusions, was independently conducted by the authors without LLM involvement in the core research process.

## A.2  NOTES FOR THE INFORMATION-FLOW VIEW

**DPI under deterministic inference.** After training, condition on fixed parameters $\theta$. Let $H = E_\theta(X)$, $O = P_\theta(H)$, $Y = \pi_\theta(O)$ be deterministic mappings of $X$. For any measurable $f$, DPI gives $I(Z; f(X)) \leq I(Z; X)$. Applied stage-wise to the composition,

$$I(Z; Y) \leq I(Z; O) \leq I(Z; H) \leq I(Z; X).$$

If inference includes independent randomness $U$ (e.g., stochastic decoding), write $Y = g(O, U)$ with $U \perp Z \mid O$, so $I(Z; Y) \leq I(Z; O, U) = I(Z; O)$.

**Chain-rule identity (for intuition).** For random variables $Z, O_1, O_2$, the chain rule gives

$$I(Z; O_1, O_2) = I(Z; O_1) + I(Z; O_2 \mid O_1) \geq I(Z; O_1).$$

We use this only to express complementarity succinctly; no quantitative claim is made.

## A.3  ALGORITHM FOR TIME-AWARE COVERAGE-WEIGHTED LINEAR INTERPOLATION

We upsample the aggregated CED features $\mathbf{H}_{\mathrm{agg}}$ to align with the Whisper feature timeline. Naive nearest-neighbor or plain linear interpolation can introduce temporal artifacts, especially for transient sounds and near clip boundaries. Our method respects (i) the precise timestamps of each feature window and (ii) pre-computed *coverage weights* that down-weight features partially derived from zero-padding.

**Assumptions and Notation.** We denote the aggregated CED sequence as $\mathbf{H}_{\mathrm{agg}} \in \mathbb{R}^{T_c \times D}$ with per-feature centers $t_c[0], \ldots, t_c[T_c - 1]$ (monotonically increasing), and the target Whisper centers $t_w[0], \ldots, t_w[T_w - 1]$ (also monotonic). All timestamps share the same unit (e.g., mel frames or milliseconds). A small constant $\varepsilon > 0$ (we use $10^{-8}$) is added for numerical stability. When $T_c \leq 1$, we simply repeat the sole vector to length $T_w$.

**Coverage weights.** For each CED window $\ell$, its coverage weight $c[\ell] \in [0, 1]$ measures the fraction of the window overlapping valid (non-padded) audio. Concretely, if a window starts at $\mathrm{start}_\ell$ and ends at $\mathrm{end}_\ell$ with window size $t_{\mathrm{sz}}$, and the valid audio spans $[0, T_{\mathrm{mel}} - 1]$, then

$$c[\ell] \;=\; \frac{\max\big(0,\, \min(\mathrm{end}_\ell,\, T_{\mathrm{mel}} - 1) - \mathrm{start}_\ell + 1\big)}{t_{\mathrm{sz}}} \,.$$

Thus $c = 1$ for fully valid windows and decreases as padding overlap grows.

**Target centers for Whisper.** Let $step_{mel}$ be the mel frames per Whisper token and $center_{mel}$ its center offset. We use

$$t_w[k] \;=\; k \cdot step_{mel} \;+\; center_{mel}, \qquad k = 0, \ldots, T_w - 1. \tag{6}$$

In our implementation, $step_{mel} = 8$ and $center_{mel} = 4$.

**Discussion.** By reweighting both neighbors with $c[\ell]$ in the numerator *and* renormalizing by the weighted sum in the denominator, features largely sourced from padded/silent regions contribute less to the aligned representation, especially near boundaries. In practice we use a vectorized implementation (single search-sorted, broad-casted arithmetic) that avoids Python loops while preserving the above semantics.

---

**Algorithm 1** Time-Aware, Coverage-Weighted Linear Interpolation (0-based indexing)

---

**Require:** Aggregated CED features $\mathbf{H}_{\mathrm{agg}} \in \mathbb{R}^{T_c \times D}$
**Require:** CED centers $t_c[0..T_c - 1]$ (monotonic), Whisper centers $t_w[0..T_w - 1]$ (monotonic)
**Require:** Coverage weights $c[0..T_c - 1]$, with $c[\ell] \in [0, 1]$
**Require:** Stability constant $\varepsilon > 0$ (e.g., $10^{-8}$)
**Ensure:** Aligned features $\mathbf{H}_{\mathrm{aligned}} \in \mathbb{R}^{T_w \times D}$
 1: Initialize $\mathbf{H}_{\mathrm{aligned}}$ as a tensor of shape $(T_w, D)$
 2: **if** $T_c \leq 1$ **then**
 3:     **return** $\mathbf{H}_{\mathrm{aligned}} \leftarrow$ repeat the sole vector to length $T_w$
 4: **end if**
 5: **for** $k \leftarrow 0$ to $T_w - 1$ **do**
 6:                                      $\triangleright$ Locate neighbors of $t_w[k]$ in $t_c$ via binary search
 7:     $r \leftarrow \mathrm{searchsorted}(t_c, t_w[k])$
 8:     $r \leftarrow \mathrm{clamp}(r, 1, T_c - 1), \quad l \leftarrow r - 1$
 9:                                  $\triangleright$ Linear interpolation factor with clamping
10:     $\alpha \leftarrow \dfrac{t_w[k] - t_c[l]}{t_c[r] - t_c[l] + \varepsilon}; \quad \alpha \leftarrow \mathrm{clamp}(\alpha, 0, 1)$
11:                           $\triangleright$ Coverage-weighted, normalized interpolation
12:     $\mathbf{x}_L, \mathbf{x}_R \leftarrow \mathbf{H}_{\mathrm{agg}}[l], \mathbf{H}_{\mathrm{agg}}[r]$
13:     $c_L, c_R \leftarrow c[l], c[r]$
14:     $\mathrm{num} \leftarrow (1 - \alpha)(c_L \mathbf{x}_L) + \alpha(c_R \mathbf{x}_R)$
15:     $\mathrm{den} \leftarrow (1 - \alpha) c_L + \alpha c_R + \varepsilon$
16:     $\mathbf{H}_{\mathrm{aligned}}[k] \leftarrow \mathrm{num}/\mathrm{den}$
17: **end for**
18: **return** $\mathbf{H}_{\mathrm{aligned}}$

---

## A.4 TRAINING SETTING

Table 4: Training hyperparameters. Encoders (Whisper, CED-Base) are frozen throughout.

| | Stage 1 (Alignment) | Stage 2 (SFT, LoRA) |
|---|---|---|
| Trainable modules | CED Aggregator | CED Aggregator, Whisper adapter; LLM via LoRA |
| Dataset | EvA-Alignment | EvA-Perception |
| Epochs | 5 | 2 |
| Per-device batch | 2 | 2 |
| Grad. accumulation | 8 | 16 |
| Global batch (8×A100) | 128 | 256 |
| Optimizer | AdamW ($\beta_2$=0.95, wd 0.1) | AdamW ($\beta_2$=0.95, wd 0.1) |
| LR / schedule / warmup | $1 \times 10^{-3}$ / cosine / 1% | $5 \times 10^{-5}$ / cosine / 1% |
| Max seq length | 512 | 1024 |
| LoRA (LLM) | – | $r$=64, $\alpha$=64, dropout = 0.05; targets=q,k,v,o (include_mlp=False) |
| Modules to save | | model.vq_adaptor |
| Checkpoint export | split every epoch (keep last 5) | split every epoch (keep last 5) |
| Distributed setup | torchrun, 8×A100-80GB | torchrun, 8×A100-80GB |
| DeepSpeed ZeRO-3 | prepared (config available), *off* by default | prepared, *off* by default |
| Runtime (wall-clock) | ~12h | ~12h |

## A.5 DATA CONSTRUCTION

---

**Instruction for Caption Generation**

**Rigorous Multimodal Information Integration and Pure Audio Description Expert**
**Core Task**
You are an expert specializing in audio information processing. Your goal is: to integrate and analyze textual descriptions from multiple modalities, strictly controlling cross-modal interference, and to perform cross-reasoning and correction. Ultimately, you should generate a **purely audio-focused**, temporally ordered, accurate, and detailed description of the audio content in fluent English, while marking potential ambiguities that are **only based on auditory perception**. **Absolutely forbidden:** including any visual information, specific speech transcript content, or ambiguity annotations derived from audio-video inconsistencies in the final output.
**Input Sources (may contain errors, hallucinations, or incompleteness):**

- **Audio Tags:** A set of frame-level sound category labels annotated by humans. These represent the most prominent acoustical features perceptible to humans, with **high reliability**. Format: `[start time, end time, event]`, e.g., `[start time: 9.0, end time: 10.0, event: Generic impact sounds]`.

- **Audio Description:** A textual description of the audio content (may include sound events, environmental sounds, music, human voice characteristics, etc.). This is an **important basis** for describing audio facts and must be cross-validated with tags and music description.

- **Speech Content (ASR):** Automatic speech recognition results. This is used **only** to confirm the existence of human voices, judge non-content features (e.g., speech vs. nonverbal sounds, emotional tone), and assist in inferring possible scenes or events. **The specific text content must never appear in the final output** (not quoted, summarized, or paraphrased). If empty, it means no obvious human voice or only non-speech behaviors (e.g., breathing, crying, background chatter).

- **Music Description:** Contains information about music elements (features, instruments, rhythm, etc.) and other scene-related sounds. **Musical features are highly reliable. If empty, it means no clear music.** Non-musical descriptions here are lower priority and secondary to audio tags, audio descriptions, and ASR.

- **Video Description:** Visual scene description. Used **only under strict conditions** (see Step 2 "Positive Correction") to disambiguate uncertain auditory sources and identify inconsistencies between hearing and vision. **Never** speculate or describe sound sources, positions, or visual actions based solely on video. If empty, no visual assistance is available.

**Processing Steps:**

1. **Multimodal Parsing:** Extract key sound events, source characteristics, environment, and music elements from each source. Give priority to audio tags. From ASR, only detect voice presence and non-content features, possibly aiding environment/emotion inference, but never include speech text itself.

2. **Auditory Fact Determination and Cross-modal Correction:**
   - Base facts primarily on: **Audio Tags > Music Description (music part) ≈ Audio Description > Speech presence (ASR) > Non-music part of Music Description**.
   - Apply **Positive Correction with video** only if audio information is ambiguous and video provides clear, reasonable confirmation of the specific sound source (e.g., generic rumble corrected to airplane noise if airplane is explicitly shown). If tags already specify the type, no correction applies.
   - If video does not support or contradicts, never override auditory facts; only internally mark inconsistency for conservative phrasing later.
   - Adopt extreme conservatism when conflicts remain unresolved: omit or cautiously phrase uncertain elements.

3. **Pure Auditory Ambiguity Inference:** List ambiguities only from hearing, such as similarity of sounds (e.g., car vs. plane), multiple possible sources, or common perceptual misinterpretations. **Never include visual-based ambiguities.**

4. **Reliability Assessment:** If audio facts are extremely scarce, or sources are severely conflicting and cannot yield reliable auditory facts, directly output: UNCERTAIN_AUDIO_INFORMATION_DETECTED.

5. **Final Audio Caption Generation:** If reliable:
   - Generate a fluent, precise, audio-focused English description that preserves event order, number of occurrences, auditory features (sound type, timbre, rhythm, loudness, space, etc.).
   - Integrate confirmed auditory facts, using cautious wording for uncertain elements ("sounds like", "a sound resembling ... is heard", "potentially ...").
   - Strictly exclude: visual details, speech text, or speculation not supported by input sources.
   - Express emotional tone if strongly supported by audio (e.g., voice emotion or music mood).

**Output Format:** Normally, output must be JSON:

```
{
  "Potential ambiguities": [
    "Ambiguity description 1 based solely on auditory perception.",
    "Ambiguity description 2 based solely on auditory perception."
  ],
  "Audio caption": "Final audio description focusing solely on audible elements and their
  auditory characteristics, detailed and fluent English. Use conservative language when
  uncertain."
}
```

In the special case of Step 4 failure, output only: UNCERTAIN_AUDIO_INFORMATION_DETECTED.

Figure 3: Instruction for Caption Generation.

---

**Instruction for QA Generation**

You are a data generator that converts a single audio caption into QA pairs. Write as if you actually listened to the audio.

**Hard rules:**

1. **Grounding:** Use **only** the given description as grounding. Do not invent facts beyond what the description supports. Do not produce ASR-style verbatim transcripts.
2. **Output format:** Return JSON Lines (JSONL). Each line is one JSON object. No markdown, no backticks, no extra prose, no blank lines.
3. **JSON validity:** Use straight ASCII quotes. No trailing commas. All keys required. Types must match the schema.
4. **Brevity:** Closed-ended answers must be short (yes/no, true/false, a label, a small set, or a number) or MCQ (A–D, answer is a single letter).
5. **Evidence:** support_span must not be empty. Provide exact words/phrases taken from the description that justify the answer. Do not add commentary.
6. **Difficulty tagging:** Use the boolean field ishard. Set it to true **only** for the most difficult items in the batch; otherwise false.
7. **No meta-reference to source:** In both question and answer, never mention the existence of any "caption/description/text". Write from an *audio-listener* perspective, not a reader perspective.

**Schema (each JSON object must match):**

```
{
  "question": string,
  "answer": string,
  "answer_style": "close" | "open",
  "type": one of [
    "presence", "counting", "temporal_order", "concurrency", "trend",
    "comparative", "scene_spatial", "causality", "mood_style",
    "submodality_decomposition", "music_feature", "speech_structure",
    "cross_modal_integration", "higher_order_semantics",
    "aesthetic_function", "scene_inference",
    "rhythm_structure", "tension_dynamics"
  ],
  "submodality": array of ["speech","music","sound"],
  "support_span": array of strings (never empty),
  "confidence": number in [0,1],
  "ishard": boolean
}
```

**Closed-ended conventions:**

- Yes/No answers must be exactly `"Yes"` or `"No"`.
- True/False answers must be exactly `"True"` or `"False"`.
- MCQ: include the options concisely in the question; set answer to exactly one letter among `"A"`,`"B"`,`"C"`,`"D"`.
- Counts are integers as strings (e.g., `"2"`).
- Options or sets should be concise strings (e.g., `"drums"`, `"drums, guitar"`).

**Design goals:**

- Favor integrative, cross-modality questions that combine speech/music/sound cues.
- Include deep semantic understanding: structural roles, function vs. ornament, implied emotion arcs, rhythm-harmony interplay, tension build/release.
- Keep questions diverse (no near-duplicates) and tightly grounded by the description's content—yet never *mention* the description explicitly.

Figure 4: Instruction for QA Generation.

Table 5: Composition of EvA-Perception and EvA-Alignment datasets.

| Dataset | Constituent Sources | Modality | Quantity |
|---|---|---|---|
| EvA-Alignment | EvA-Captions | Sound, Speech, Music | 53,934 |
| | AudioTime (Xie et al., 2025a) | Sound | 5,000 |
| | CommonVoice (Xie et al., 2025a) | Speech | 20,000 |
| | MusicBench (Melechovsky et al., 2023) | Music | 30,000 |
| | MusicCaps (Agostinelli et al., 2023) | Music | 4,852 |
| Total | | | 113,786 |
| EvA-Perception | EvA-Captions | Sound, Speech, Music | 53,934 |
| | EvA-QA | Sound, Speech, Music | 525,673 |
| | AudioSkills: Counting-QA (Goel et al., 2025) | Sound | 46,266 |
| | ESC50 (Piczak, 2015) | Sound | 2,000 |
| | AudioTime (Xie et al., 2025a) | Sound | 5,000 |
| | DCASE2025_T5 (Yang et al., 2025) | Sound | 10,687 |
| | CommonVoice (Xie et al., 2025a) | Speech | 20,000 |
| | MusicBench (Melechovsky et al., 2023) | Music | 30,000 |
| | MusicCaps (Agostinelli et al., 2023) | Music | 4,852 |
| | AudioSkills: MagnaTagATune (Goel et al., 2025) | Music | 364,760 |
| Total | | | 1,063,172 |

## A.6 CASES IN EVA-CAPTIONS AND EVA-QA

---

**Case1**

Captions:
The audio opens with a lively live performance featuring rhythmic electric guitars, prominent bass tones, punchy percussion, **and** a mellow synth undertone, all layered with sustained crowd energy. A male vocalist delivers a passionate vocal passage that blends with the driving instrumentation **before** transitioning into instrumental segments. A high-pitched female vocalist emerges briefly, **adding** melodic contrast to the arrangement. The track culminates in an abrupt crescendo of an intense, unified crowd shout that escalates sharply **before** terminating abruptly, leaving a sudden silence.

QA pairs:
Q: What is the primary emotional effect created by the abrupt final shout and silence?
A: A jarring sense of closure with heightened tension release

Q: How many distinct vocalists are present in the audio?
A: 2

Q: What type of vocal presence is heard after the initial male vocalist's passage?
A: A high-pitched female vocalist

Q: What is the primary emotional effect created by the abrupt final shout and silence?
A: A jarring sense of closure with heightened tension release

---

Figure 5: Instruction for QA Generation.

## A.7 INTRODUCTION ON BENCHMARKS

**MMAU** covers broad audio modalities with a mixture of perceptual, information-extraction, and reasoning questions; **MMAR** emphasizes multi-step inference across hierarchical layers; **MMSU** targets spoken language understanding, including fine-grained linguistic and paralinguistic phenomena. **Reasoning**. Here, *MMSU*'s native tags are kept, *MMAU*'s *information extraction*/*reasoning* map to Perception/Reasoning, and *MMAR*'s *Signal+Perception*/*Semantic+Culture* likewise.

We discover that, in the original MMAU test set, the reference answers are distributed unevenly across choice positions: **A: 39.5%, B: 27.1%, C: 20.8%, and D: 12.6%.** Such imbalance may bias

Case1

Captions:
Persistent wind noise dominates throughout the recording with intermittent radio transmissions containing male speech. The radio communication **begins** abruptly with fragmented phrases mentioning 'Blackbird,' **followed** by three distinct mechanical ticks at approximately the same temporal proximity. A sharp burst of static noise momentarily **interrupts** the transmission **before** resuming with the male voice reiterating 'Blackbird' amidst continuing wind interference. The radio remains active **throughout the recording** with overlapping wind noise maintaining consistent background presence.

QA pairs:
Q: Which sound consistently overlays the entire recording?
A: Wind noise

Q: Is there a sharp burst of static noise present in the audio?
A: Yes

Q: How does the recurrence of 'Blackbird' contribute to the overall atmosphere of tension?
A: It creates a sense of urgency and repetition amid unstable communication, suggesting unresolved or critical information being transmitted through persistent interference.

Figure 6: Instruction for QA Generation.

evaluation for models with positional preferences (e.g., favoring earlier options). To mitigate this artifact, we randomized the order of choices, ensuring that the final distribution of correct answers is balanced across positions. All reported MMAU results in the main paper are based on this balanced setting.

Table 6: Main results on MMAU-mini-test.

| Model | Sound | Speech | Music | Avg. |
|---|---|---|---|---|
| Qwen2-Audio | 58.86 | 47.75 | 44.31 | 50.30 |
| Qwen2.5-Omni | 73.87 | 65.47 | 67.96 | 69.10 |
| Kimi-Audio | 74.77 | 62.35 | 64.24 | 67.19 |
| Audio-Reasoner | 65.77 | 66.07 | 66.77 | 65.00 |
| R1-AQA | 74.47 | 65.17 | 66.77 | 68.80 |
| **EvA(Ours)** | **80.78** | **68.47** | **74.65** | **74.63** |

Table 7: Main results on MMAR.

| Model | Single Modality | | | Mixed Modality | | | | Avg. |
|---|---|---|---|---|---|---|---|---|
| | Sound | Speech | Music | S-M | S-S | M-S | S-M-S | |
| Qwen2-Audio | 52.73 | 42.86 | 34.95 | 36.36 | 50.46 | 45.12 | 50.00 | 44.80 |
| Qwen2.5-Omni | 59.39 | 61.22 | 48.06 | 54.55 | 61.01 | **64.63** | 58.33 | 58.30 |
| Kimi-Audio | 55.76 | 59.86 | 45.15 | 36.36 | 61.01 | 54.88 | 45.83 | 55.40 |
| Audio-Reasoner | 50.30 | 49.66 | 38.35 | 36.36 | 56.42 | 48.78 | 50.00 | 48.70 |
| R1-AQA | **60.00** | 51.36 | 42.23 | 54.55 | 57.80 | 52.44 | 45.83 | 52.30 |
| **EvA(Ours)** | 55.76 | **63.01** | **50.25** | **63.64** | **63.76** | 57.32 | **79.17** | **59.30** |

## A.8 ADDITIONAL ANALYSIS: FREQUENCY-BAND ABLATION OF THE CED PATH

To better understand how EvA exploits spectral cues, we perform an exploratory ablation on the CED branch by masking coarse frequency bands at inference time. The CED encoder partition the 64 Mel bins into four contiguous groups and treat them as approximate 2 kHz bands: $[0, 2), [2, 4), [4, 6), [6, 8)$ kHz. On top of the frozen CED encoder, we insert a band mask before

Table 8: Main results on various audio benchmarks. All numbers are average scores.

| Setting | MMAU | MMAR | MMSU | CochlScene | TUT2017 | VocalSound |
|---------|------|------|------|------------|---------|------------|
| mask 0-2kHz | 72.20 | 55.53 | 63.72 | 60.67 | 44.62 | 90.11 |
| mask 2-4kHz | 72.30 | 55.63 | 63.72 | 60.91 | 46.07 | 89.97 |
| mask 4-6kHz | 72.30 | 56.24 | 63.82 | 60.61 | 45.64 | 90.17 |
| mask 6-8kHz | 72.40 | 55.33 | 63.88 | 61.00 | 45.47 | 90.17 |
| mask 0-4kHz | 72.30 | 54.93 | 63.11 | 60.67 | 45.30 | 90.06 |
| mask 4-8kHz | 71.90 | 55.43 | 63.98 | 60.78 | 46.50 | 90.23 |
| left 0-2kHz | 72.10 | 55.73 | 63.34 | 60.34 | 46.15 | 89.95 |
| left 2-4kHz | 72.50 | 54.23 | 63.52 | 60.41 | 44.70 | 90.03 |
| left 4-6kHz | 71.90 | 54.12 | 63.38 | 60.02 | 44.87 | 90.25 |
| left 6-8kHz | 72.80 | 55.33 | 62.81 | 60.27 | 44.27 | 89.78 |
| **use 0-8kHz(EvA)** | **73.90** | **59.76** | **62.24** | **74.94** | **66.24** | **93.48** |

the Aggregator: for each configuration, a binary vector of length four determines which bands are zeroed out and which are kept. All experiments share the same single-encoder EvA backbone; we only change the band mask at inference without re-training.

Table 8 reports the results on multi-task benchmarks (MMAU, MMAR, MMSU) and three specialized perception tasks (CochlScene, TUT2017, VocalSound). We consider both "*mask X*" (drop one or two bands and keep the others) and "*left X*" (keep only one band and mask the rest), and compare them with the full 0–8 kHz setting used in EvA.

**Broadband cues are consistently better than any subset.** Across all benchmarks, any masking or single-band configuration yields lower scores than using the full 0–8 kHz range. The full-band EvA remains the best-performing setting in all cases, which supports the interpretation that EvA relies on broadband, complementary spectral information rather than a single dominant frequency region.

**Acoustic scene and event classification benchmarks show mild sensitivity to low frequencies.** For CochlScene and TUT2017, masking the lowest band (0–2 kHz) tends to be among the weaker configurations, and keeping only 0–2 kHz is also not the best single-band setting. Together, these observations suggest that low-frequency components provide useful contextual cues for ambient scenes, but are most effective when combined with mid- and high-band information. We view this as a mild trend rather than a strong claim, since the absolute differences between band configurations remain relatively small.

**Vocal nonverbal behavior tasks exhibit no strongly preferred band.** On VocalSound, the task about vocal nonverbal behavior, all band-masked settings lie within a narrow range and no specific 2 kHz band clearly stands out as uniquely critical. Using only the highest band (6–8 kHz) tends to be slightly weaker than using lower bands, but the gaps are again small compared with the overall gain from enabling the full 0–8 kHz CED path.

Similarly, the aggregated MMAU/MMAR/MMSU scores are designed to combine heterogeneous tasks, making them less sensitive to any single frequency band; here the main takeaway is simply that broadband CED fusion is more reliable than any restricted band subset. They corroborate our main conclusion that EvA benefits from broadband, multi-band CED features, and we do not observe evidence that the model's performance is dominated by a single narrow frequency region.

A.9  QUALITATIVE ANALYSIS

We illustrate differences using AudioCaps case studies (Fig. 7). Compared to Qwen2.5-Omni and Kimi-Audio, **EvA** produces captions that are both faithful and more expressive. In Case 1, EvA captures sequential tone shifts (calm speech → child excitement → laughter), while baselines misinterpret events. In Case 2, EvA details prosody and temporal dynamics (speech rhythm, two distinct

> **Case1: AudioCaps/103394.wav**
> **Human Reference:**
>    "A man is speaking followed by a child speaking and then laughter",
>    "A man speaking as a man talks over an intercom in the background followed by a girl talking then a group of people laughing and a man and woman talk"
>
> **Qwen2.5-Omni:** man and child speaking
> **Kimi-Audio:** a man speaks and a baby cries
> **EvA (Ours): A man speaks calmly and clearly, followed by a child speaking excitedly. A group of people laugh, and the man speaks again** in a friendly tone

> **Case2: AudioCaps/103975.wav**
> **Human Reference:**
>    "A woman speaks and continues to do so as a dog starts barking"
>
> **Kimi-Audio:** a woman speaks and a dog barks
> **Qwen2.5-Omni:** A woman is speaking and a dog is barking.
> **EvA (Ours): A woman speaks clearly and calmly,** her voice steady and unhurried. She **pauses briefly** before **speaking again. A dog barks twice,** the first bark **short and sharp,** the second **longer and more insistent.**

Figure 7: Qualitative comparison of captions generated by different models on AudioCaps examples.

dog barks), going beyond the coarse outputs of others. These examples show how EvA's evidence-first fusion enhances both accuracy and richness in captioning.

## A.10 STRUCTURAL COMPARISON WITH Q-FORMER–BASED FUSION

In this appendix, we clarify the structural differences between EvA's CED Aggregator and a Q-Former–based fusion scheme (as instantiated by our SALMONN-style variant), and relate them to the temporal behavior illustrated in Figure 8.

**Token length and temporal granularity.**    Figure 8 compares the audio token sequence length after feature fusion in SALMONN versus EvA. The Q-Former in SALMONN maps a long sequence of encoder features to a much shorter set of latent queries, thereby introducing temporal compression. In contrast, EvA preserves full temporal resolution: Time-Aware Alignment produces $H_{\text{aligned}}$ on the Whisper timeline, and the inject-and-add fusion in Eq. (6) yields $E_{\text{fused}}$ without reducing sequence length. This non-compressive design ensures that short transient events and fine-grained temporal structure remain available to the downstream LLM, which is particularly important for perception-heavy benchmarks.

**Access to acoustic evidence.**    Q-Former architectures typically only consume the encoder's final-layer features. For audio encoders, these top-layer representations are often semantically collapsed and may lose low- and mid-level cues that are crucial for environmental sound and event recognition. EvA introduces an explicit cross-layer bypass that aggregates multiple CED layers, $H_4, H_8, H_L$, into $H_{\text{agg}}$ via the two-stage cascaded cross-attention. This allows the fusion module to reuse shallow, mid-level, and high-level acoustic information instead of relying solely on the last encoder layer.

**Interaction mechanism.**    Q-Former modules rely on a bank of static learnable queries that are shared across all inputs: the same latent queries attend to encoder features regardless of the specific audio content. In contrast, EvA performs content-adaptive cross-layer retrieval inside the CED path: $H_L$ first attends to $H_8$, and the resulting representation then attends to $H_4$, forming a top–down hierarchy $H_L \to H_8 \to H_4$. This hierarchical attention enables the model to selectively recover fine-grained evidence from lower layers conditioned on the current high-level context, which is not possible when only the final encoder layer is exposed to a fixed query set.

**Summary.**    Taken together, these structural differences—multi-layer access, content-adaptive cross-layer retrieval, and non-compressive temporal fusion—provide a different inductive bias from Q-Former–based designs. This is consistent with our ablations in Table 3, where the SALMONN-

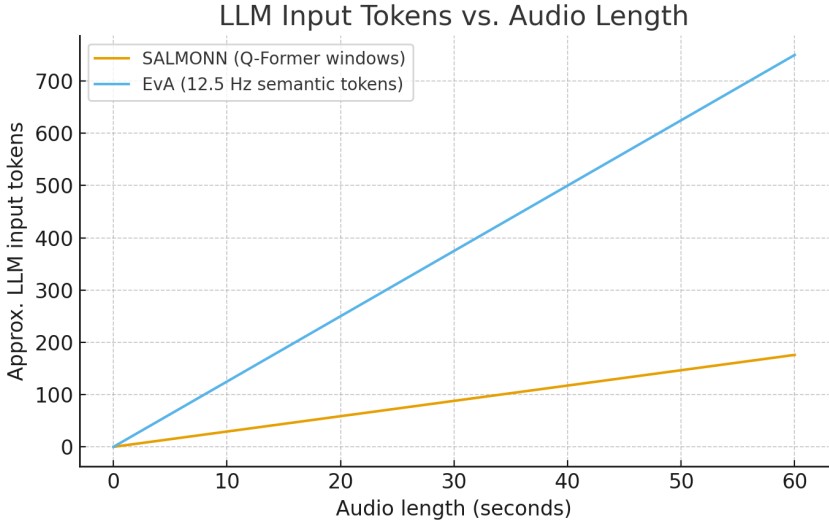

Figure 8: Comparison of audio token sequence length after feature fusion. SALMONN's Q-Former compresses audio tokens into a shorter latent sequence, while EvA preserves sequence length via inject-and-add fusion.

style Q-Former variant improves over weaker baselines but still underperforms EvA's hierarchical Aggregator on perception benchmarks.