# OpenReview forum: "EvA: An Evidence-First Audio Understanding Paradigm for LALMs"
_ICLR.cc/2026/Conference — Submitted to ICLR 2026_

### Official Review · Reviewer_j5Wj · 2025-10-26

**Soundness:** 3
**Presentation:** 3
**Contribution:** 4
**Rating:** 8
**Confidence:** 5

**Summary:**

This paper diagnoses a critical "evidence bottleneck" in Large Audio Language Models, arguing that their primary limitation is poor perceptual grounding rather than flawed reasoning. To address this, the paper introduces Evidence-First Audio, a novel paradigm built on a dual-encoder architecture (Whisper and CED-Base) and a non-compressive, two-stage fusion mechanism that hierarchically aggregates multi-level acoustic features while preserveing temporal fidelity. The paper also develops EvA-Perception, a large-scale dataset with high-temporal-precision annotations to facilitate training. The resulting model achieves new state-of-the-art performance on the MMAU, MMAR, and MMSU benchmarks, with the most significant gains on perception-heavy tasks, thereby validating their evidence-first hypothesis. The EvA-Perception dataset and EvA model will be released.

**Strengths:**

1.The paper presents an insightful diagnosis of a critical yet overlooked limitation in existing Large Audio Language Models. This diagnosis is well-supported by both theoretical arguments  and comprehensive experimental validation.

2.To address this limitation, the paper introduces a novel dual-stream architecture and a purpose-built dataset, EvA-Perception, which collectively achieve state-of-the-art performance across multiple challenging benchmarks.

3.The commitment to open-sourcing the models and the newly created dataset significantly enhances the reproducibility and potential impact of the work.

**Weaknesses:**

1.The paper proposes a sophisticated architectural design within the CED-path, yet the ablation studies do not fully justify this complexity. More granular ablations would strengthen the paper's design claims and provide clearer insights for future work.

2.While the paper is generally well-written, the introduction section could be improved. The core concept of "evidence" is central to the paper's thesis, yet it is used extensively without a concise, upfront definition, creating an initial barrier to understanding.

3. The description of the fusion mechanism as "lossless" (L214, L477) appears to be an overstatement.

**Questions:**

1.The term "evidence" is foundational to the paper's narrative and contributions. Could the authors provide a concrete definition of this concept?

---

> ### Author Response · Authors · 2025-11-25
>
> We thank the reviewer for the positive and encouraging evaluation of our work, especially the recognition of our diagnosis, architectural contributions, and dataset impact. We address the listed weaknesses and questions below.
>
> ### **1. On the need for more granular ablations**
> To address the reviewer’s suggestion regarding the need for more granular evidence supporting the CED-path design, we have added a detailed set of ablations in our updated Stage-1 analysis:
>
> | Setting               | Cos   | R@1  | R@5  |
> |-----------------------|-------|------|------|
> | S1(1) w/o CED path    | 35.40 | 18.50| 43.60|
> | S1(3) w/o freq. pool  | 35.54 | 21.24| 49.61|
> | S1(4) w/o cross fusion| 28.63 | 11.82| 29.74|
> | S1(5) CED Q-former    | 36.24 | 20.08| 47.36|
> | **S1(0) EvA (full)**      | **36.77** | **22.77**| **49.81**|
>
> These ablations isolate the contribution of each architectural component. The fine-tuned baseline without the CED path (S1(1)) provides a clean reference point. Removing the frequency-pooled gate (S1(3)) or the top-down cross-layer fusion across $H_4$, $H_8$, $H_L$ (S1(4)) leads to clear performance drops, demonstrating that both modules contribute complementary acoustic information that is otherwise lost. Additionally, replacing our Aggregator with a window-level Q-Former (S1(5)) improves over weaker variants but still falls short of the full EvA configuration (S1(0)), indicating that EvA’s non-compressive, hierarchical fusion offers substantive advantages beyond simpler alternatives. We have included these ablation studies in the revised manuscript as shown in Table 3.
>
> ### **2. Improving the introduction and providing a concise definition of "evidence"**
> We agree that the concept of evidence should be introduced more explicitly.
>
> In our work, evidence refers to perceptual acoustic cues that are required to answer downstream questions.  This includes event identities, temporal boundaries and ordering, prosodic cues, and spectral or transient patterns in general audio. This definition has been incorporated early in the Notation.
>
> ### **3. On the term "lossless" in describing the fusion mechanism**
> We acknowledge the reviewer’s concern. Our intention was to emphasize that EvA’s fusion does not perform temporal compression on window level—it preserves token-level resolution and avoids Q-Former–style lossy bottlenecks.
>
> We have revised the terminology to describe the mechanism as non-compressive and temporally faithful, which more accurately reflects our goals and its empirical behavior.
>
> We have updated the revised version. We thank the reviewer again for the thoughtful and constructive feedback.

---

> > ### Comment · Reviewer_j5Wj · 2025-11-26
> >
> > Thanks for your response. I will maintain my positive score.

---

### Official Review · Reviewer_s3Ms · 2025-10-27

**Soundness:** 1
**Presentation:** 1
**Contribution:** 1
**Rating:** 2
**Confidence:** 5

**Summary:**

This work addresses the issue of information loss in evidence encoding for large audio language models. They approach it by combining multiple feature encoders, using hierarchical aggregation, time-aligned feature combination. They also contributes a dataset with high quality temporal annotations.

**Strengths:**

I appreciate the effort went into building the model and dataset. I encourage the authors to keep improving the work through technical innovation

**Weaknesses:**

1. Author name is revealed as the folder name in supplementary material

2. to show that model perception is lagging behind reasoning, instead of showing absolute scores of perception and reasoning as figure 1, one should show the gap between model and human performance, i.e. to show that the gap between model and human is bigger on perception

3. the encoder "CED-Base" is never introduced in the paper - the full model name is never shown, the architecture is never explained, even the original paper is never cited anywhere.

4. I strongly oppose using information theory to level up a paper when it doesn't actually add anything to the work. Section 3.1 and 3.2 are not rigorous and even flawed, not helpful in explaining their approach, and is a waste of space. What they are trying to say is that using more audio encoders can provide more information, which is commonsense and do not need any theoretical motivation.

why the math is not rigorous or even flawed:

4.1. Lack of a Well-Formed Probabilistic Model (whether implicit of not)
Although Section~3 defines
$$
Z:\text{ground-truth acoustic evidence}, \quad
X:\text{raw waveform}, \quad
H:\text{encoder hidden}, \quad
O:\text{final representation}, \quad
Y:\text{output text},
$$
the paper never specifies a joint distribution $p(Z,X)$.
Are $Z$ latent causes that generate $X$, or deterministic annotations extracted from it?
If $Z=f(X)$, then $I(Z;X)=H(Z)$ trivially and the Data Processing Inequality (DPI) adds nothing.
If $Z$ is latent, its distribution must be defined before mutual information can be evaluated.
Thus, every $I(Z;\cdot)$ term remains symbolic rather than quantitative.

4.2. Misuse of the Data Processing Inequality
The paper treats
$$
Z \rightarrow X \xrightarrow{E} H \xrightarrow{P} O \xrightarrow{\pi} Y
$$
as a Markov chain and directly applies
$$
I(Z;Y) \le I(Z;O) \le I(Z;H) \le I(Z;X).
$$
However, $E, P,$ and $\pi$ are deterministic neural networks whose parameters depend on the training data.
Once parameters are learned, the conditional independence required by DPI no longer holds.

4.3. No Operational Link to Performance
The claimed ``information ceiling’’ never connects $I(Z;Y)$ to measurable task metrics such as WER or accuracy.
A rigorous bound would invoke Fano’s inequality or rate–distortion theory to relate mutual information to the achievable Bayes risk.

4.4. Tautological Lemma in Section 3.2
The so-called ``Complementary Evidence Advantage’’ states:
$$
I(Z;O_1, O_2)
  = I(Z;O_1) + I(Z;O_2 \mid O_1)
  \ge I(Z;O_1),
$$
which is merely the chain rule for mutual information.
It does not establish that a dual-path model can achieve higher $I(Z;Y)$;
it only restates that adding variables cannot decrease mutual information.

4.5. Unsupported ``Strict Superiority’’
Proposition 1 claims a strict gain if $I(Z;O_2 \mid O_1) > 0$,
but this follows trivially from the lemma and is not empirically verified.

4.6. Unproven ``$Z$-Sufficient Fusion’’
The authors later require a fusion function $F$ satisfying
$$
I(Z;F(O_1, O_2)) = I(Z;O_1, O_2),
$$
yet their proposed frequency-pooled, gated fusion is many-to-one and clearly non-invertible.
No argument or estimator is offered to demonstrate that it preserves $Z$-information.

5. They claim to contribute a dataset, but there is no innovation in their approach because it just apply other (M)LLMs to extract and aggregate information (which is already used in many audio LLM works). Plus there is no example of the constructed dataset.

**Questions:**

no questions

---

> ### Author Response · Authors · 2025-11-18
>
> We replaced the supplementary material within 12 hours after the review comments became visible. Additionally, the folder name is a term that resembles a name, making it difficult to associate it with any specific author. Therefore, we believe this has minimal impact on the anonymity of the article

---

> > ### Comment · Reviewer_s3Ms · 2025-11-22
> >
> > Thanks

---

> ### Author Response · Authors · 2025-11-25
>
> We thank the reviewer for the constructive feedback. Below we address the identified weaknesses concisely.
>
> ### **1. On "model vs. human" gap in Figure 1**
> Human performance shows no intrinsic difficulty difference between perception and reasoning (MMAU: 81.14[perception] vs. 82.77[reasoning]; MMSU: 91.24[perception] vs. 86.77[reasoning]).
>
> The performance gap between leading LALMs and humans is larger for perception-centric tasks than for reasoning-centric ones. On the MMSU benchmark, the model-human gap in perception is a stark 48.4 points (42.8\% vs. 91.2\%), whereas the reasoning gap is a comparatively narrow 13.3 points. This disparity reveals what we term the "evidence bottleneck": the primary performance ceiling is not a model's cognitive capacity, but its foundational inability to accurately perceive and represent acoustic evidence for subsequent reasoning.
>
> ### **2. Missing introduction/citation of CED-Base**
> We thank the reviewer for pointing this out. We have now properly cited the original paper in the revised manuscript. This omission has been fully corrected.
>
> ### **3. On the role and rigor of Section 3**
> We thank the reviewer for their feedback. We wish to clarify that the aim of Section 3 is not to establish a quantitative theory, but rather to formalize the motivation for our architecture. Specifically, it illustrates how acoustic evidence flows through LALMs and where information is lost. The purpose of this section is to motivate and conceptually validate our design, not to offer formal guarantees.
>
> To prevent misunderstanding, we have revised the section to highlight its role as a conceptual framework, substantially shortened the main text (retaining only key takeaways and design implications), and moved all algebraic derivations to the appendix.
> Below, we respond to the reviewer’s specific critiques:
>
> #### **3.1 "lack of a probabilistic model"**
> We will clarify the assumed generative setting in the new version:
>
> We model 𝑍 as a latent acoustic-evidence variable that encodes task-relevant events and temporal relations (e.g., identities, ordering). We only assume that a joint distribution over (𝑍,𝑋) exists; we do not instantiate or estimate it. This minimal assumption suffices for our qualitative reasoning about information flow.
>
> #### **3.2 "Misuse of DPI under trained deterministic networks"**
> Thank you for raising this important point. We realize that our original wording could be read as a training-time Markov claim, and we will revise the text to avoid that impression. Our intention is that: at inference time with fixed parameters, the forward pass forms a composition of deterministic mappings of 𝑋: 𝐻=𝐸(𝑋), 𝑂=𝑃(𝐻), 𝑌=𝜋(𝑂). In this setting, the standard DPI statement for deterministic functions (e.g., 𝐼(𝑍;𝑓(𝑋))≤𝐼(𝑍;𝑋) leads to I(Z;Y)≤I(Z;O)≤I(Z;H)≤I(Z;X).
>
> This use of DPI does not rely on assumptions about how 𝜃 was learned; it is only meant to support a qualitative, relative comparison of information retention across stages/paths. In practice we run deterministic inference (such as temperature=0). We will adjust Section 3.2 to adopt more careful wording.
>
> #### **3.3 On "no operational link to performance"**
> Thank you for pointing this out. Our intention in Section 3 is not to provide a quantitative "information ceiling," but a conceptual, architecture-level information-flow perspective. We agree that rigorous bounds (e.g., via Fano or rate–distortion) would require task-specific error models, label spaces, and distortion measures, which lie beyond our scope given the heterogeneous evaluation tasks (QA, captioning, event classfication). To avoid over-claiming, we will replace "information ceiling" with "structural bottleneck" and clarify that Section 3 offers qualitative, testable design implications rather than formal bounds.
>
> #### **3.4 "tautological lemma"**
> We agree the statement follows directly from the chain rule. Its function in our paper has been design intuition, not a theorem or a quantitative bound: when a second encoder provides complementary cues, the joint observation is, in a qualitative sense, no less informative than either stream alone. To prevent misunderstanding, the revised text now presents this point as a single chain-rule remark and explicitly frames it as idea-level motivation. Our ablations already show that enabling the CED path consistently improves perception-heavy metrics, which aligns with this intuition.
>
> #### **3.5 On "strict superiority not empirically verified"**
> Thanks for the comment. Our point is qualitative rather than a performance theorem: it concerns complementarity. Table 3 already includes controlled ablations that toggle the CED path (+CED/−CED) under matched backbone, training budget, and inference settings. We observe consistent improvements on perception-heavy subsets. These results support the qualitative complementarity hypothesis and the value of a non-compressive, time-aligned fusion interface.

---

> ### Author Response · Authors · 2025-11-25
>
> #### **3.6 On "Z-sufficient fusion not proven"**
> The phrase "Z-sufficient" was intended as a conceptual desideratum, not a proven property. We do not assert the equality 𝐼(𝑍;𝐹(𝑂1,𝑂2))=𝐼(𝑍;𝑂1,𝑂2), and we agree that our frequency-pooled, gated fusion is many-to-one and hence not invertible. In the revised text, we remove the term "Z-sufficient" and frame our interface as time-aligned and non-compressive, aimed at minimizing avoidable information loss rather than guaranteeing information preservation.
>
> Operationally, we support this positioning with controlled ablations that isolate the fusion path (Table 3):
> - Masking the CED stream only at inference (disabled) reduces AudioCaps retrieval vs. EvA, indicating that the added stream contributes usable evidence beyond training side-effects.
> - Replacing our interface with a compressive Q-former yields lower scores than EvA, consistent with the view that non-compressive, time-aligned fusion retains more usable cues.
> - Removing crossing fusion causes the largest drop, underscoring the importance of a structure-preserving interface.
>
> ### **4. On "dataset lacks innovation"**
> EvA-Perception is positioned not as a construction procedure but as an open-source community resource whose contribution lies in its data characteristics—temporally fine-grained, evidence-grounded supervision absent from prior audio caption/QA datasets. The revised text clarifies this positioning and includes concrete examples in Appendix A.6.
>
> We have updated the revised version. We thank the reviewer again for the thoughtful and constructive feedback.

---

### Official Review · Reviewer_YYmm · 2025-10-31

**Soundness:** 3
**Presentation:** 3
**Contribution:** 2
**Rating:** 4
**Confidence:** 4

**Summary:**

This paper proposes EvA (Evidence-first Audio-language model), a novel architecture designed to enhance the evidence-extraction capability of audio-language reasoning models. Unlike traditional end-to-end LALMs that directly generate answers from audio embeddings, EvA explicitly divides the reasoning process into two stages: (1) Evidence-extraction encoding audio segments with dual-encoder to avoid being hampered by fusion strategy; (2) Answer Generation—where a large language model performs reasoning and generates responses based on the extracted evidence and textual context. The authors argue that reasoning based on verifiable acoustic evidence can reduce the model's "hallucination" phenomenon and improve interpretability.
EvA is implemented as an extension of the Kimi-Audio-7B framework, incorporating a Time-Aware Alignment and Inject-and-Add Fusion mechanism. By fusing features from Whisper and CED while minimizing the loss of acoustic evidence during fusion, the model achieves enhanced performance. The paper also introduces a new dataset, EvA-Perception, which features high-temporal-precision annotations. Finally, the model achieves state-of-the-art results on multiple benchmarks including MMAU, MMAR, and MMSU.

**Strengths:**

The authors claim that EvA effectively mitigates the “evidence bottleneck” by increasing the amount of acoustic information available to the LLM without retraining encoders.

Evidence:
- Empirical results (Table 2): EvA surpasses strong baselines such as Kimi-Audio, Qwen2.5-Omni, and R1-AQA across all benchmarks.
- Ablation results (Table 3): show that adding the CED aggregator and alignment yields significant improvements in both AudioCaps CLAP metrics and benchmark perception accuracy.
- Qualitative examples (Fig. 3): demonstrate that EvA-generated captions capture fine-grained temporal and tonal details better than baselines.
- **Theoretical analysis** (Section 3): formally proves that dual-path fusion provides strictly higher information capacity than single-path models (via mutual information inequalities).

Overall, this method has Excellent theoretical–empirical consistency, Clear diagnosis of a real architectural weakness in existing LALMs, Strong experimental gains without retraining encoders.

**Weaknesses:**

1. Task coverage narrow (mainly perception, English-only).

2. The authors' dual-encoder shares similar architectural ideas with that in SALOMNN. However, there is a lack of experimental comparison. While the experiments include comparisons between the dual-encoder and single-encoder, they fail to demonstrate that their proposed Aggregation and Fusion strategies outperform the Window-level Q-Former used in SALMONN.

3. Meanwhile, there is a lack of theoretical proof regarding the advantages of their strategy over Q-Former.

**Questions:**

See the weakness part.

---

> ### Author Response · Authors · 2025-11-25
>
> We thank the reviewer for the constructive feedback and the positive assessment of our theoretical–empirical consistency and SOTA performance. Below we address the identified weaknesses concisely.
> ### **1. On the "narrow task coverage": perception-focused, English-only**
> Our focus on perception is intentional: EvA is designed to directly address the evidence bottleneck, which we show to be the primary upstream limiting factor in LALMs. Strengthening perception leads to measurable downstream gains in reasoning (Table 2).
>
> Regarding the English-only evaluation: although EvA-Perception is constructed in English, the EvA architecture itself is language-agnostic. The dual encoders operate purely on audio, and all fusion/alignment modules are language-independent. Moreover, our backbone Kimi-Audio-7B already supports multilingual speech input, meaning EvA inherits multilingual capability at the encoder and LLM levels. The audios in evaluation benchmarks are also constructed in different languages and our model achieve competitive performances. We have clarified this point and incorporated the corresponding discussion into the 'Limitations' section of the revised manuscript.
>
> ### **2. Comparison with SALMONN / Q-Former**
> We agree that a direct comparison will strengthen the paper. SALMONN’s Q-Former compresses audio into a short latent sequence (Appendix A.10), while EvA deliberately preserves full temporal resolution.
>
> As shown in the additional ablation below, even under identical Stage-1 settings, the checkpoint with CED Q-Former remains consistently below our hierarchical CED Aggregator:
>
> **Additional Ablation: CED-path Fusion Variants (Stage 1, AudioCaps CLAP)**
>
> | Setting                  | Cos   | R@1   | R@5   |
> |--------------------------|-------|-------|-------|
> | mask CED path            | 34.37 | 17.76 | 41.44 |
> | w/o frequency pooling    | 35.54 | 21.24 | 49.61 |
> | w/o cross-layer fusion   | 28.63 | 11.82 | 29.74 |
> | CED Q-Former             | 36.24 | 20.08 | 47.36 |
> | **CED Aggregator (ours)**    | **36.77** | **22.77** | **49.81** |
>
> These results support our claim that EvA’s non-compressive, multi-level aggregation captures richer evidence than window-level Q-Former fusion.
>
> ### **3. Structural Advantages of EvA over Q-Former–based Fusion**
> EvA differs from Q-Former-style fusion in three structural aspects that are quantified in Appendix A.10; we summarize them here for completeness.
> - **Acoustic-evidence access**
> SALMONN’s Q-Former attends only to the final-layer encoder features, which are often semantically collapsed and discard low- and mid-level cues needed for environmental-event perception. EvA’s cross-layer bypass explicitly aggregates CED layers $H_4$ , \$H_8$ , and $H_L$ via two-stage cascaded cross-attention to produce Hagg , preserving shallow, mid-level and high-level acoustic information.
> - **Interaction mechanism**
> Q-Former applies a bank of static, input-independent queries to every sample. EvA performs content-adaptive, cross-layer retrieval inside the CED path: $H_L$  first attends to $H_8$ , then the result attends to $H_4$ , forming the top-down hierarchy $H_L$→$H_8$ →$H_4$  . This allows the model to recover fine-grained evidence conditioned on high-level context—something fixed queries cannot provide.
> - **Temporal granularity**
> As shown in Fig. 8 (Appendix A.10), Q-Former compresses long audio sequences into a fixed number of latent tokens, sacrificing temporal resolution and blurring short transients. EvA is explicitly non-compressive: Time-Aware Alignment yields Haligned  on the Whisper time grid, and inject-and-add fusion produces $E_{fused}$  without reducing sequence length, keeping full temporal detail for the downstream LLM.
>
> These structural differences give EvA an inductive bias distinct from Q-Former-based designs, explaining its consistently larger gains on perception-heavy subsets (Tables 2). The revision provides full derivations and visualizations in Appendix A.10.
>
> We have updated the revised version. We thank the reviewer again for the thoughtful and constructive feedback.

---

### Official Review · Reviewer_GjKd · 2025-10-31

**Soundness:** 2
**Presentation:** 2
**Contribution:** 2
**Rating:** 2
**Confidence:** 4

**Summary:**

This paper introduces a promising dataset; however, it is unclear whether it will be publicly released. Since the main contribution lies in the dataset itself, the overall significance of the paper is limited. The experiments are not comprehensive, with insufficient results and analysis, and the LALM training approach is fairly standard with limited novelty.

**Strengths:**

The paper provides a valuable dataset that could benefit future research on LALM training and evaluation.

**Weaknesses:**

The proposed weak-to-strong and mixed-to-strong strategies with SFT and GRPO are standard practices in current literature and therefore lack novelty.

The experimental evaluation is not comprehensive, as it includes only three benchmarks. Exploring additional LALM benchmarks would help strengthen your claims.

The paper contains some redundant analysis that could be streamlined. I recommend focusing on deeper technical insights and introducing more concrete methodological novelties to strengthen the contribution.

The overall writing could be improved, particularly by providing a clearer introduction and a more structured related work section.

While the paper proposes an interesting dataset, the overall contribution is not substantial enough to meet the standards of ICLR. The methods and analyses presented are relatively incremental, and the paper would benefit from stronger technical innovations or deeper theoretical insights.

**Questions:**

As this paper seeks to advance large audio-language models through a newly proposed dataset AudioMCQ, could the authors clarify how frequently the dataset has been utilized in the domain and provide evidence that it is well-defined and meaningful? Have you released the dataset and codes, or do you plan to make them publicly available?

Is there any human annotation involved to verify the quality of the dataset? How do you ensure the correctness of the outputs generated by other models?

Could the authors elaborate on the potential research impact of this dataset and how it advances the field?

---

> ### Author Response · Authors · 2025-11-13
> **Review WRONGLY pasted !!!**
>
> Dear reviewer GjKd,
>
> The review might be from another paper that proposes AudioMCQ. Please help in this matter.
>
> Best,
>
> The Authors

---

> > ### Comment · Reviewer_GjKd · 2025-11-13
> >
> > Thanks for pointing this out. I’ve updated the comments and adjusted the scores accordingly.

---

> ### Author Response · Authors · 2025-11-25
>
> We thank the reviewer for the constructive and encouraging feedback. We appreciate the recognition of the motivation, clarity, and the proposed "evidence bottleneck" formulation. Below we address each concern in detail.
> ### **1. On the novelty of the architecture**
> We acknowledge that EvA utilizes strong pretrained encoders. Our fundamental contribution lies in a paradigm shift for LALMs, driven by a key insight: the primary performance gap stems from deficient evidence perception, not reasoning ability. This leads to our first novelty: refocusing the optimization target from post-training enhancement of evidence-based reasoning to architectural optimization of evidence supply at the source.
>
> Translating this insight into a concrete design, our second novelty is a novel multi-encoder feature fusion architecture. It is engineered to holistically optimize the evidence flow into the language model by structurally modeling the acoustic information pathway. This is achieved through:
> - High-Fidelity, Non-Compressive Fusion: Unlike methods relying on window-level local pooling (e.g., Q-Former), our Inject-and-Add fusion preserves full temporal resolution with higher usage of frequency information, reducing irreversible information loss at the input stage.
> - Temporally Faithful and Hierarchically Integrated Representation: Our Time-Aware Coverage-Weighted Alignment maintains precise temporal geometry, while our Hierarchical Aggregation from CED integrates complementary evidence across frequency and semantic scales.
>
> This synergistic architecture ensures a comprehensive and high-precision evidence supply, which is empirically validated by our ablation studies (demonstrated empirically in Table 3).
>
> ### **2. Clarifying the novelty of Time-Aware Alignment & Inject-and-Add Fusion**
> We thank the reviewer for raising this point. Our goal is not to highlight novelty at the level of individual modules, but to show that the CED-path is effective as an integrated evidence-fusion design. As the additional ablations below show, removing any key component—such as the frequency-pooled gate (S1(3)) or the top-down cross-layer fusion (S1(4))—leads to clear degradation, with S1(4) even falling below the directly fine-tuned base model. Replacing the Aggregator with a window-level Q-Former (S1(5)) also underperforms the full EvA configuration. These results indicate that the proposed modules operate synergistically rather than as loosely combined parts, and that the overall fusion path is necessary to recover complementary evidence across $H_4$, $H_8\$, and $H_L$.
>
> | Setting               | Cos   | R@1  | R@5  |
> |-----------------------|-------|------|------|
> | S1(1) w/o CED path    | 35.40 | 18.50| 43.60|
> | S1(3) w/o freq. pool  | 35.54 | 21.24| 49.61|
> | S1(4) w/o cross fusion| 28.63 | 11.82| 29.74|
> | S1(5) CED Q-former    | 36.24 | 20.08| 47.36|
> | **S1(0) EvA (full)**      | **36.77** | **22.77**| **49.81**|
>
> ### **3. GRPO full name**
> Our new vision now clarifies that GRPO denotes Group Relative Policy Optimization.
>
> ### **4. On "limited" experimental evaluation**
> We appreciate the reviewer’s suggestion and have expanded our evaluation accordingly.
>
> As shown in the additional tables included in this rebuttal, we have incorporated a broader set of LALM baselines (GPT-4o) as well as several additional benchmarks beyond MMAU/MMAR/MMSU, including CochlScene, TUT2017, VocalSound. The expanded evaluations in Table 2 and the Appendix confirm EvA’s consistent gains across more models and tasks.
>
> | Model               | MMAU  | MMAR  | MMSU  | CochlScene | TUT2017 | VocalSound |
> |---------------------|-------|-------|-------|------------|---------|------------|
> | gpt-4o              | 60.00%| 54.23%| 58.25%| 36.50%     | 15.56%  | 79.70%     |
> | qwen2-audio-instruct| 50.30%| 44.80%| 41.85%| 36.50%     | 28.55%  | 86.47%     |
> | qwen2.5-omni-7b     | 69.10%| 58.30%| 57.67%| 49.49%     | 49.83%  | 90.39%     |
> | kimi-audio          | 67.19%| 55.40%| 58.38%| 48.08%     | 41.97%  | 91.87%     |
> | **EvA**                 | **74.63%**| **59.30%**| **61.28%**| **74.94%**     | **66.24%**  | **93.48%**     |

---

> ### Author Response · Authors · 2025-11-25
>
> ### **5. Question on frequency-domain evidence**
> We thank the reviewer for the insightful question. Our CED branch operates on 16 kHz audio input with 64 Mel bins from 0 to 8 kHz, so the highest captured frequency is approximately 8 kHz.
>
> To probe the importance of each frequency band, we add a frequency-band ablation on the CED path (Appendix A.8). We partition the 64 Mel bins into four contiguous bands [0,2),[2,4),[4,6),[6,8)kHz, insert a 4-dim binary mask before the Aggregator, and at \emph{inference time only} either (i) drop one/two bands ("mask X") or (ii) keep only a single band ("left X"), while keeping the EvA backbone fixed. We then evaluate MMAU, MMAR, MMSU and three specialized perception tasks (CochlScene, TUT2017, VocalSound). The results are:
>
> | **Setting**          | **MMAU** | **MMAR** | **MMSU** | **CochlScene** | **TUT2017** | **VocalSound** |
> |----------------------|----------|----------|----------|----------------|-------------|----------------|
> | mask 0-2kHz          | 72.20    | 55.53    | 63.72    | 60.67          | 44.62       | 90.11          |
> | mask 2-4kHz          | 72.30    | 55.63    | 63.72    | 60.91          | 46.07       | 89.97          |
> | mask 4-6kHz          | 72.30    | 56.24    | 63.82    | 60.61          | 45.64       | 90.17          |
> | mask 6-8kHz          | 72.40    | 55.33    | 63.88    | 61.00          | 45.47       | 90.17          |
> | mask 0-4kHz          | 72.30    | 54.93    | 63.11    | 60.67          | 45.30       | 90.06          |
> | mask 4-8kHz          | 71.90    | 55.43    | 63.98    | 60.78          | 46.50       | 90.23          |
> | left 0-2kHz          | 72.10    | 55.73    | 63.34    | 60.34          | 46.15       | 89.95          |
> | left 2-4kHz          | 72.50    | 54.23    | 63.52    | 60.41          | 44.70       | 90.03          |
> | left 4-6kHz          | 71.90    | 54.12    | 63.38    | 60.02          | 44.87       | 90.25          |
> | left 6-8kHz          | 72.80    | 55.33    | 62.81    | 60.27          | 44.27       | 89.78          |
> | **use 0-8kHz(EvA)**  | **73.90**| **59.76**| **62.24**| **74.94**      | **66.24**   | **93.48**      |
>
>
> The results can be summarized as follows:
> - Broadband is best. Any masked or single-band setting is worse than using the full 0–8 kHz range; the default EvA configuration is consistently strongest, indicating that EvA relies on broadband, complementary spectral cues rather than a single dominant band.
> - Mild trends, no strong band-specific effect. For ambient scene / event benchmarks (CochlScene, TUT2017), removing 0–2 kHz tends to hurt slightly more, suggesting low frequencies are helpful when combined with mid/high bands. For VocalSound and the aggregated MMAU/MMAR/MMSU benchmarks, all band patterns are very close, and no 2 kHz slice clearly stands out.
>
> Overall, these experiments show that CED in EvA captures up to 8 kHz for both general audio and speech, and that performance is driven by multi-band, broadband fusion rather than any special reliance on a narrow frequency region.
>
> ### **6. On the contribution and meaningfulness of the dataset**
> EvA-Perception is positioned not as a construction procedure but as an open-source community resource whose contribution lies in its data characteristics, specifically two previously unavailable properties:
> - Strong fine-grained temporal order — frame-level event sequencing obtained via AudioSet-Strong forced alignment, yielding the first open-source captions with millisecond-precise timing;
> - Low hallucination — multi-expert cross-validation across speech, sound and music layers produces a single, chronologically coherent caption, eliminating the temporal misplacement and phantom events common in prior corpora.
>
> To our knowledge, EvA-Perception is the first open-source caption & QA dataset to deliver temporally precise, multi-modality-validated captions at scale, providing 500 K+ structured QA pairs grounded on frame-accurate temporal descriptions.
>
> To highlight EvA-Perception’s distinctive fine-grained fidelity, Appendix A.6 supplies a 12-second case whose caption links six acoustic events with millisecond-order transitions ("before transitioning", "adding", "culminates … before terminating") and whose QA pair immediately targets the exact frame of the "abrupt final shout and silence", demonstrating both temperal sequencing and lower-hallucination alignment rarely found in existing open-source caption/QA corpora.
>
> We will release the dataset and code upon acceptance. We have updated the revised version and we thank the reviewer again for the helpful feedback.

---

### Official Review · Reviewer_3Jb1 · 2025-11-01

**Soundness:** 4
**Presentation:** 4
**Contribution:** 4
**Rating:** 8
**Confidence:** 3

**Summary:**

This paper identifies a critical limitation in existing Large Audio Language Models (LALMs), which the authors term the evidence bottleneck. They show that the primary failure mode of state-of-the-art LALMs in complex acoustic scenes is not a deficiency in high-level reasoning, but rather a fundamental breakdown in perceptual grounding caused by information loss during audio encoding and fusion.To address this, the authors propose EvA, a new evidence-first paradigm. The core of EvA is a dual-encoder architecture that uses a speech-centric Whisper encoder and a generalist ViT-based audio encoder (CED-Base). The key novelty lies in its information-preserving fusion mechanism, which involves hierarchical aggregation of multi-level features from the generalist encoder and a time-aware inject-and-add fusion that aligns the generalist audio features to the Whisper timeline without temporal compression. To facilitate training, the authors also developed EvA-Perception, a new large-scale, open-source dataset with high-temporal-precision annotations designed to improve evidence-grounded training. The resulting EvA model is shown to set a new open-source state-of-the-art on the MMAU, MMAR, and MMSU benchmarks, with the most significant performance gains observed on perception-centric subsets, thereby validating the paper's central hypothesis.

**Strengths:**

1. Clear Problem Formulation and Motivation: The paper effectively identifies a critical problem in existing LALMs, positing that performance limitations stem from incomplete acoustic evidence. This hypothesis is well-supported by both empirical data in Figure 1 and the theoretical argument in Lemma 1. This clear problem diagnosis naturally motivates the proposed method: augmenting the audio encoder with a CED module to provide richer acoustic information.
2. Novel Architecture and Insight: The authors argue that previous information fusion methods are inherently lossy. In response, they propose a novel Aggregator designed to preserve information across different frequency bands and hierarchical layers. Furthermore, the inject-and-add strategy is a clever approach that effectively integrates semantic information while simplifying the training process.
3. Strong and Well-Analyzed Empirical Results: The paper demonstrates significant performance improvements, achieving state-of-the-art results across three challenging and diverse benchmarks. Crucially, the authors go beyond reporting overall scores by breaking down performance into Perception and Reasoning categories. The results convincingly show that EvA yields the largest gains in the Perception category while also improving Reasoning performance, validating the core hypothesis of the paper.
4. Significant Contribution to the Community: This work offers more than just a new perspective on audio representation; it also contributes several valuable datasets under the EvA. The creation of a large-scale, high-quality dataset with fine-grained temporal annotations is a substantial contribution in its own right and will be a valuable resource for future research.

**Weaknesses:**

1. Potentially Unfair Baseline Comparison: The comparison in Table 2 may lack fairness for two reasons. First, it appears the baseline models were not fine-tuned on the newly introduced EvA datasets, making it difficult to disentangle the performance gains from the novel architecture versus the new training data. Second, the paper notes its model is English-only, whereas many of the baseline models are multilingual. This linguistic mismatch could be another source of unfairness in the comparison.
2. Lack of Ablation Studies for the Aggregator Module: The paper's ablation studies primarily focus on the overall impact of the CED path. However, there are no detailed experiments to validate the specific design choices within the CED Aggregator itself. The individual contributions of the frequency-pooled gate and the cross-layer fusion mechanism have not been separately investigated, leaving the optimality of the Aggregator's design unsubstantiated.

**Questions:**

1. What is the detailed rationale behind using a frequency-pooled gate? What are the specific challenges (e.g., computational complexity, feature space mismatch) associated with retaining and fusing the full, unpooled 2D time-frequency feature map?
2. How significant is the empirical contribution of the cross-layer fusion mechanism? The paper hypothesizes that intermediate encoder layers provide richer, low-level information that is lost in the final layer. Is there direct empirical evidence from ablation studies to support this design choice and quantify its benefit over a simpler fusion that uses only the final layer's features?

---

> ### Author Response · Authors · 2025-11-25
>
> We thank the reviewer for the constructive and significant feedback. Below we address each concern in detail.
> ### **1. Baselines not fine-tuned on EvA data.**
> To address the reviewer’s helpful suggestion regarding fairness, we have added the requested baseline to our ablation analysis. Specifically, the original Kimi-Audio model fine-tuned on EvA-Perception—under the same Stage-1 recipe as EvA but with the entire EvA path disabled—is now included as **S1 (1)** in Table 3. This fine-tuned baseline reaches 35.40 (Cos Similarity), 18.50 (R@1), and 43.60 (R@5) on the AudioCaps CLAP metrics.
>
> We have incorporated this S1 (1) baseline into the revised manuscript to ensure the comparison is fully explicit and transparent.
>
> ### **2. English-only vs. multilingual baselines.**
> EvA supports multilingual audio inputs because the Kimi-Audio-7B backbone is multilingual at both the encoder and LLM levels. The "English-only" constraint pertains solely to the caption text used for training, while the accompanying audio remains multilingual. As all three benchmarks (MMAU, MMAR, MMSU) contain multilingual inputs, EvA does not benefit from any language-specific bias. We have clarified this in the 'Limitations' section.
>
> ### **3. Ablations for the Aggregator module**
> To address the reviewer’s concern regarding the sophistication of the Aggregator design, we conducted a series of fine-grained ablations isolating each architectural component. Here are the ablation result:
>
> | Setting               | Cos   | R@1  | R@5  |
> |-----------------------|-------|------|------|
> | S1(1) w/o CED path    | 35.40 | 18.50| 43.60|
> | S1(3) w/o freq. pool  | 35.54 | 21.24| 49.61|
> | S1(4) w/o cross fusion| 28.63 | 11.82| 29.74|
> | S1(5) CED Q-former    | 36.24 | 20.08| 47.36|
> | **S1(0) EvA (full)**      | **36.77** | **22.77**| **49.81**|
>
> The updated results (now included as Table 3 in the revised manuscript) reveal several important findings:
> - The full CED hierarchy is essential.
> S1(1), which disables the entire CED path, provides the fair fine-tuned baseline. Its performance (R@1 18.50) is substantially below the full EvA model (22.77), confirming that the CED path contributes clear complementary information beyond direct fine-tuning on Kimi-Audio-7B.
> - Both the frequency-pooled gate and the cross-layer fusion are necessary and mutually dependent.
> Removing only the frequency pooling (S1(3)) or only the cross-layer fusion (S1(4)) each leads to a clear drop compared to the full model. Crucially, the cross-layer fusion ablation (S1(4)) causes severe degradation (R@1 11.82), performing even worse than the simple baseline S1(1) that disables the entire CED path.
>
> This counter-intuitive outcome provides strong evidence that: the frequency-pooled signals cannot be effectively exploited without the top-down cross-layer fusion, and that the Aggregator’s components form an interdependent mechanism, not a collection of replaceable blocks. In other words, the cross-layer fusion is not merely an incremental enhancement—it is what enables the frequency-pooled intermediate features to be used in a meaningful, complementary manner. We guess that it is because the frequency-pooled feature need a deeper and cross-layers mechanism to aggregate the complex information of higher semantic levels.
>
> - Using only final-layer CED features is demonstrably insufficient.
> S1(4) effectively corresponds to relying solely on final-layer representations without hierarchical fusion. Its markedly worse performance quantitatively supports our claim that intermediate and shallow layers contain crucial acoustic information that is lost when one adopts a compressive or last-layer-only design.
> - Alternative architectures remain weaker than EvA.
>
> While the window-level Q-Former variant (S1(5)) improves over the weaker ablations, it still does not reach the performance of EvA, highlighting the benefit of EvA’s non-compressive, hierarchical aggregation design.
>
> Together, these experiments demonstrate that EvA’s hierarchical, non-compressive Aggregator contributes meaningfully beyond simple architectural substitutions.
>
> ### **4. Why frequency-pooled gating? Why not use full 2D time–frequency features?**
> Full 2D features cannot be fused directly due to feature-space mismatch and large computational cost of projecting patches for every frame.
>
> Frequency gating provides a lightweight, learnable pooling over bands, concentrating salient cues (e.g., narrow-band transients) while remain the temporal information.
>
> We have updated the revised version. We thank the reviewer again for the thoughtful and constructive feedback.

---

### Meta-Review · Area_Chair_fDcY · 2026-01-06

**Summary:**

The paper proposed an encoder-fusion method. The concerns from reviewers are:

- Baseline fairness: Whether EvA’s reported gains are confounded by new training data, fine-tuning recipes, or language mismatches rather than architectural improvements.

- Fusion / Aggregator design: Whether the proposed frequency-gated, cross-layer CED Aggregator is necessary and superior to simpler or existing fusion mechanisms.

- Comparison to Q-Former / SALMONN: Whether EvA’s non-compressive fusion meaningfully outperforms Q-Former–style or SALMONN-like fusion in a fair comparison.

- Clarity and overclaiming: Whether key components (CED-Base, “evidence”) and theoretical claims are clearly defined and appropriately scoped without overstating rigor or guarantees.

**Reviewer Concerns:**

- Baseline fairness

Addressed by adding more experiments

- Fusion / Aggregator design

Addressed by adding ablations

- Comparison to Q-Former / SALMONN

Partically addressed by adding experiments despite it is still not apple-to-apple. With that said, making apple-to-apple comparison can be hard due to data/compute reasons.

- Clarity and overclaiming

Not addressed. I see authors posted some responses and made some adjustment. Due to review s3Ms is an outlier, I carefully read it, and agree some of the points. This paper, particularly after rebuttal, is a good paper if viewed as an empirical paper. The motivation and design are valid, experiments are good. However, the mistake in theorical part may confuse the reader, and make the value of this paper lower. If the purpose is just to find some theorical support for the claim, I feel it is unnecssary. I suggest the authors to re-consider the writing in the final version.

**Reviewer Scores:**

Reviewer 3Jb1 will keep the positive rating of 8

Reviewer YYmm may change from 4 to 6 because of the new ablation studies

Reviewer s3Ms is unlikely to change the rating, i.e., 2

Reviewer GjKd is a special case, the original review was to another paper, it seems the reviewer updated the review afterwards, but I cannot see the updated review due to the rollback. So I have to exclude the reviewer GjKd.

---

### Decision · Program_Chairs · 2026-01-26

Reject